# Aging and sperm signals alter DNA break formation and repair in the *C. elegans* germline

Erik Toraason[☉], Victoria L. Adler[ID][☉], Diana E. Libuda[ID]*

University of Oregon, Department of Biology, Institute of Molecular Biology, Eugene, Oregon, United States of America

☉ These authors contributed equally to this work.

* dlibuda@uoregon.edu

**Data Availability Statement:** All relevant data are within the manuscript and its Supporting Information files.

**Funding:** This work was supported by the National Institutes of Health T32GM007413 and

## Abstract

Female reproductive aging is associated with decreased oocyte quality and fertility. The nematode *Caenorhabditis elegans* is a powerful system for understanding the biology of aging and exhibits age-related reproductive defects that are analogous to those observed in many mammals, including dysregulation of DNA repair. *C. elegans* germline function is influenced simultaneously by both reproductive aging and signals triggered by limited supplies of sperm, which are depleted over chronological time. To delineate the causes of DNA repair defects in aged *C. elegans* germlines, we assessed both DNA double strand break (DSB) induction and repair during meiotic prophase I progression in aged germlines which were depleted of self-sperm, mated, or never exposed to sperm. We find that germline DSB induction is dramatically reduced only in hermaphrodites which have exhausted their endogenous sperm, suggesting that a signal due specifically to sperm depletion downregulates DSB formation. We also find that DSB repair is delayed in aged germlines regardless of whether hermaphrodites had either a reduction in sperm supply or an inability to endogenously produce sperm. These results demonstrate that in contrast to DSB induction, DSB repair defects are a feature of *C. elegans* reproductive aging independent of sperm presence. Finally, we demonstrate that the E2 ubiquitin-conjugating enzyme variant UEV-2 is required for efficient DSB repair specifically in young germlines, implicating UEV-2 in the regulation of DNA repair during reproductive aging. In summary, our study demonstrates that DNA repair defects are a feature of *C. elegans* reproductive aging and uncovers parallel mechanisms regulating efficient DSB formation in the germline.

## Author summary

Aging leads to a decline in the quality of the female reproductive cells, known as oocytes. Oocytes subjected to reproductive aging experience an increase in both infertility and aneuploidies that cause miscarriages and birth defects. The nematode *Caenorhabditis elegans* is a classic model system used to determine the mechanisms of aging. Old *C. elegans* oocytes accrue many defects which may contribute to their reduced quality, including

Achievement Rewards for College Scientists (ARCS) Foundation Award to E.T. and National Institute of General Medical Sciences (R35GM128890) to D.E.L. D.E.L. is also a recipient of a March of Dimes Basil O'Connor Starter Scholar award and Searle Scholar Award. The funders had no role in the study design, data collection and analysis, decision to publish, or preparation of the manuscript.

**Competing interests:** The authors have declared that no competing interests exist.

dysregulation of DNA repair. *C. elegans* fertility and germline function are also regulated oocyte-independently by sperm-dependent signals. To determine how aging and sperm may independently impact DNA repair in aging *C. elegans* oocytes, we control oocyte aging and sperm presence independently to evaluate their effects on DNA break formation and repair. We find that running out of sperm reduces the levels of DNA breaks which are produced, but the efficiency of DNA repair declines during aging, independent of sperm effects. We also identify a protein which specifically promotes DNA repair in the oocytes of young animals, suggesting that this protein may regulate DNA repair in the germline during aging. Taken together, our research defines aging-specific and aging-independent mechanisms which regulate the genome integrity of oocytes.

## Introduction

Genome integrity must be preserved during gamete development, as any genetic defects incurred may have detrimental effects on progeny or fertility. Meiosis, the specialized cell division that generates haploid gametes such as eggs and sperm, utilizes specific DNA repair pathways to both ensure accurate chromosome segregation and preserve genomic integrity. During early meiotic prophase I, DNA double-strand breaks (DSBs) are intentionally induced across the genome by the conserved topoisomerase-like protein Spo11 [1,2]. A specific subset of these breaks must be repaired by recombination as crossovers, creating the physical connections between homologous chromosomes required for accurate chromosome segregation. Failure to repair meiotic DSBs accurately and efficiently can contribute to infertility or risk the formation of *de novo* germline mutations.

Gamete quality is negatively impacted in organisms of advanced chronological age [3]. In many organisms, oocyte quality in particular declines starkly with maternal age [4–6]. Oocyte aging is associated with conserved phenotypic changes, including loss of sister chromatid cohesion, dysregulation of DNA repair gene expression, and derepression of heterochromatin and retroviral elements [4,7–10].

The nematode *Caenorhabditis elegans* is a key model system for the study of aging biology, including age-related infertility [11]. *C. elegans* hermaphrodites (which produce oocytes as adults) undergo reproductive senescence due to declining oocyte quality and incur many of the defects observed in the aging mammalian ovary [4,5,10,12]. Unlike many mammalian systems, however, which generate oocytes *in utero* and hold them in dictyate arrest until ovulation, *C. elegans* hermaphrodites continuously produce new oocytes during their adult reproductive period [13]. Mitotic proliferation and ovulation of oocytes is dependent upon signals from sperm, which are stored at the end of the germline in a specialized compartment called the 'spermatheca' [14,15]. "Obligate female" mutants, which do not produce sperm, therefore exhibit dramatically slowed germline proliferation and progression [14–18]. The *C. elegans* germline is organized in a spatial temporal gradient wherein oocytes mitotically proliferate at the distal tip and move proximally through the germline as they progress through meiotic prophase I [13]. Thus, oocyte nuclei at all stages of meiotic prophase I are simultaneously present in the adult germline and enable assessment of meiotic events which are dynamic across prophase, such as the induction and repair of DSBs.

Multiple lines of evidence suggest that preservation of genome integrity is important for the maintenance of oocyte quality during reproductive aging. Human females carrying DNA repair protein variants exhibit extended fertility [19]. *C. elegans* mutants with extended reproductive periods are also resilient to exogenous DNA damage and upregulate genes associated

with DNA repair [4]. Further, recent evidence demonstrated that DNA damage and repair are altered in aged *C. elegans* germlines [8,10]. By the fourth day of adulthood, *C. elegans* oocyte nuclei exhibit fewer programmed DSBs, delayed loading of recombination proteins, and increased engagement of error-prone repair mechanisms [8,10].

Sperm also regulate *C. elegans* germline physiology and reproduction. *C. elegans* hermaphrodites produce sperm only during a late stage in larval development [20]. By the third to fourth day of adulthood, these sperm are depleted, which leads to a premature cessation of reproduction in *C. elegans* hermaphrodites [4]. Sperm depletion also induces broad transcriptional remodeling independent of aging processes, resulting in a 'female-like' transcriptional profile [21]. Mating extends the hermaphrodite reproductive span on average to the sixth day of adulthood, after which declining oocyte quality limits fertility [4]. Mating and even exposure to males, however, also induces deleterious responses in hermaphrodites leading to premature demise [22,23]. It remains unknown how reproductive aging, signaling induced by the presence or depletion of sperm, and mating intersect to regulate meiotic processes in aged *C. elegans* germlines.

To define DNA repair defects which are specific to reproductive aging, we assayed levels of DSB formation and repair in the meiotic oocytes of aged mated and unmated *C. elegans* hermaphrodites, as well as feminized germline mutants that do not produce sperm (*fog-2* mutants). We demonstrate that while the depletion of sperm downregulates DSB induction in aged germlines, delayed DSB repair is a shared feature of aging germlines independent of sperm presence. Finally, we identify the E2 ubiquitin-conjugating enzyme variant UEV-2 as a putative regulator of DNA repair during germline aging. Taken together, our work distinguishes DNA repair defects specific to reproductive aging and identifies parallel mechanisms regulating gamete quality in the immortal germline.

## Methods

### *Caenorhabditis elegans* strains and maintenance

*Caenorhabditis elegans* strains were maintained at 20˚C on nematode growth medium (NGM) plates seeded with OP50 *Escherichia coli* bacteria. All experiments were performed in the N2 genetic background. Strains used in this experiment include AV761 (*GFP*::*cosa-1* II; *spo-11 (me44)* IV/ nT1[qIs51]), AV676 (*GFP*::*cosa-1* II; *fog-2(q71)* V), AV1179 (*rad-54(me98)*/tmC18 I), CB4108 (*fog-2(q71)* V*), DLW135 (*uev-2(gk960600gk429008gk429009); rgr-1(gk429013)* III), DLW199 (libIs4[*pie-1p::uev-2::unc-54 3'UTR*] III:7007600), N2 (wild type), VC30168 (Million Mutation Project strain carrying *uev-2(gk960600)*), and WBM1119 (wbmIs60 [*pie-1p::3XFLAG::dpy-10 crRNA::unc-54* 3'UTR] (III:7007600)).

Strain DLW135 was generated by backcrossing VC30168 to N2 10 times. VC30168 was created by the Million Mutations Project [24] and carried many mutations in addition to the *uev-2(gk960600)* allele of interest. Following backcrossing, mutations on Chromosomes I, II, IV, V, and X were assumed to have been eliminated. To determine the success of backcrossing on removing undesired mutations *in cis* with *uev-2* on Chromosome III, we assessed the presence of known flanking mutations to *uev-2(gk960600gk429008gk429009)*. Presence of the upstream most proximal genic mutation to *uev-2*, *pho-9(gk429005)*, was assessed via PCR amplification using OneTaq 2x Master Mix (forward primer DLO1142 5'-ACCCATTTCCCATTCAATCA-3' reverse primer DLO1143 5'-TTGTAATCTGCCCCAAAAGG-3') and subsequent HpaII restriction digest (New England Biolabs). DLW135 carried a wild type allele of *pho-9*, indicating that the region of Chromosome III upstream of *uev-2* was successfully reverted to wild-type sequence by recombination. However, the closely linked (~1 cM) downstream allele *rgr-1 (gk429013)* was preserved in DLW135, as confirmed by Sanger sequencing (Sequetech) of a

PCR amplified region of the *rgr-1* locus using OneTaq 2x Master Mix (forward primer DLO1140 5'-TGGAATGGGACTTCCTCTTG-3' reverse primer DLO1141 5'-TTTCCAA AAGCCAGGACATC-3') isolated using a GeneJET PCR Purification kit (ThermoFisher). The *rgr-1(gk429013)* allele is a single base pair substitution resulting in a S360N missense mutation. RGR-1 is a Mediator complex subunit involved in transcriptional activation that is required for embryonic viability [25]. S360N does not disrupt a predicted functional domain, and mutants carrying *rgr-1(gk429013)* survive embryogenesis and are fertile, indicating that this mutation does not severely disrupt function of the RGR-1 protein. As RGR-1 is not known to play a role in DNA damage repair, and *uev-2* has been previously demonstrated to modulate germline sensitivity to DNA damage [4], the phenotypes we observed using DLW135 in this manuscript are not best explained by the presence of the *rgr-1(gk429103)* mutation. For simplicity, DLW135 mutants are referred to as '*uev-2* mutants' in the text of this manuscript.

## Aging conditions

In experiments with aged animals, L4 hermaphrodites were isolated and maintained on NGM plates seeded with OP50 in the absence of males. Strains which produced self progeny were transferred to new NGM plates seeded with OP50 2 days post-L4 to prevent overconsumption of food from F1 progeny. At this transfer, if the experimental cohort was to be mated in order to prevent sperm depletion, young adult male N2 worms were additionally added to these plates at a ratio of ~1.5–2 males per hermaphrodite. Mated hermaphrodites were again transferred to new NGM plates with OP50 ~20–26 hours after males were added and the male animals were discarded.

For experiments in which *fog-2* worms were transiently mated once in order to assess the effects of sperm depletion on feminized germlines, *fog-2* females were transferred to a new NGM plate seeded with OP50 1 day post-L4 and young adult N2 males were also added at a ratio of ~2–3 males per female. The animals were allowed to mate for 6–8 hours, and then the females were transferred to a new NGM plate seeded with OP50 and the males were discarded. The *fog-2* mated females were then transferred to new NGM plates seeded with OP50 every ~48-60hrs to prevent their progeny from overconsuming the available food.

## CRISPR/Cas9 genome editing

Strain DLW199 was generated using the SKILODGE transgenic system [26]. WBM1119 was injected with 40ng/μL pRF4 purified plasmid, 40ng/μL purified PCR amplicon of the full *uev-2* coding sequence with 35bp homology arms to the wbmIs60 landing site (Phusion polymerase, forward primer DLO1144 5'-tcccaaacaattaaaaatcaaattttcttttccagATGCGAAGACGTAGCAA CAG-3' reverse primer DLO1154 5'-taattggacttagaagtcagaggcacgggcgcgagatgTTAGTTTTC GATGTCAATTGGT-3'), 0.25 μg/μL Cas9 enzyme (IDT), 100ng/μL tracrRNA (IDT), and 56ng/μL crRNA DLR002 (5'-GCUACCAUAGGCACCACGAG-3'). Dpy F1 progeny were isolated and screened for insertion at the wbmIs60 locus by PCR following the SKILODGE recommended protocols (primers CGSG130, CGSG117 [26]).

The candidate insertion identified among progeny from the above injected hermaphrodites contained an undesired additional 43bp of sequence between the 5' 3xFLAG tag of the edited wbmIs60 landing site and the start codon of the *uev-2* coding sequence. The strain carrying this insertion allele was backcrossed 3x to N2 and was CRISPR/Cas9 edited again to remove the undesired 5' sequence. Worms were injected with 0.25 μg/μL Cas9 (IDT), 100ng/μL tracrRNA (IDT), 28ng/μL gRNA DLR022 (5'-GAUCUUUAUAAUCACCGUCA-3'), 28ng/μL gRNA DLR023 (5'-UGUUGCUACGUCUUCGCAUC-3'), 25ng/μL ssODN donor DLO1173 (5'-AACAATTAAAAATCAAATTTTCTTTTCCAGATGCGGAGGCGAAGTAATAGACAA

TATGTTGATCTCTCATATTTTCGCGAAAC-3'), and 40ng/μL purified pRF4 plasmid. Successful removal of the 3xFLAG sequence and undesired 43bp inserted sequence were confirmed by PCR and Sanger sequencing (Sequetech).

## Nematode irradiation

*C. elegans* strains were maintained at 20˚C on NGM plates seeded with OP50 prior to and following irradiation. Irradiation was performed using a $Cs^{137}$ source (University of Oregon).

## Immunofluorescence sample preparation and microscopy

Immunofluorescence samples were prepared as in [27]. Nematodes were dissected in 1x Egg Buffer (118 mM NaCl, 48 mM $KCl_2$, 2 mM $CaCl_2$, 2 mM $MgCl_2$, 25 mM HEPES pH7.4, 0.1% Tween20) and were fixed in 1x Egg Buffer with 1% paraformaldehyde for 5 min on a Super-Frost Plus slide (VWR). Slides were then flash frozen in liquid nitrogen and the cover slip was removed before the slides were placed in ice cold methanol for 1 minute. Slides were washed in 1xPBST (1x PBS, 0.1% Tween20) 3x for 10 minutes before they were placed in Block (1xPBST with 0.7% bovine serum albumin) for a minimum of one hour. 50μL of primary antibody diluted in PBST (see below for specific antibody dilutions) was then placed on each slide and samples were incubated for 16-18hrs in a dark humidifying chamber with parafilm coverslips. Slides were then washed 3x in 1xPBST for 10 minutes. 50μL of secondary antibody diluted 1:200 in PBST were then added to each sample and slides were incubated for 2hr in a dark humidifying chamber with parafilm coverslips. Slides were washed 3x in 1xPBST for 10 minutes, and then 50μL of 2μg/mL DAPI was applied to each slide. Samples were incubated in a dark humidifying chamber with parafilm coverslips for 5 minutes, then were washed 1x in PBST for 5 minutes. Slides were mounted in VECTASHIELD mounting media (Vector laboratories) with a No 1.5 coverslip (VWR) and sealed with nail polish. All slides were maintained at 4˚C until imaging.

Immunofluorescence images were acquired at 512x512 or 1024x1024 pixel dimensions on an Applied Precision DeltaVision microscope with a 60x lens and a 1.5x optivar. All images were acquired in 3 dimensions with Z-stacks at 0.2μm intervals. In a minority of aged unirradiated germlines, we noted that most nuclei in mid-late pachytene exhibited high levels of RAD-51 and condensed DNA morphology characteristic of apoptosis. These aberrant gonads were not included in our analyses. Images were deconvolved with Applied Precision softWoRx software and individual image tiles were stitched into a single image for analysis using the Grid/Collection Stitching module in Fiji with regression threshold 0.7 [28] or using Imaris Stitcher software (Bitplane).

Specific antibodies used and their dilution factors are: Rabbit αRAD-51 (1:500), Chicken αRAD-51 (1:1000) [29], Rabbit αDSB-2 (1:5000) [30], Rabbit αGFP (1:1500) [31], Chicken αGFP(1:1000, Abcam GR236651), Alexa Fluor 488 Goat αChicken (1:200, Invitrogen A11039), Alexa Fluor 555 Goat αChicken (1:200, Invitrogen A21437), Alexa Fluor 555 Goat αRabbit (1:200, Invitrogen A21428), and Alexa Fluor 488 Goat αRabbit (1:200, Invitrogen A11008).

## Image analysis and quantification

Images were analyzed as described in [32]. Image quantification was performed using Imaris software (Bitplane). Individual nuclei within stitched gonads were identified as Surface objects (Smooth 0.1–0.15, Background 3–4, Seed Point Diameter 3–4) based on DAPI staining intensity. Manual thresholding of specific values were used per gonad to generate surfaces which represented the nuclei observed. Defined surfaces were then split to designate individual nuclei

using the Imaris Surfaces Split module. Nuclei which were partially imaged or overlapped with other nuclei were eliminated from the analysis. We previously demonstrated that a minority nuclei (~23%) in the total population must be eliminated in this manner and that the inclusion of multiple germlines in a dataset enables thorough sampling of nuclei across the course of prophase I for analysis [32]. RAD-51 foci were defined as Spot objects (Estimated XY Diameter 0.1, Model PSF-elongation 1.37, Background Subtraction enabled). To determine the number of RAD-51 foci per nucleus, we either utilized the "Find Spots Close to Surface" MATLAB module (Threshold value 0.1) or utilized the "Closest Distance to Surface" statistic calculated by Imaris to find the number of Spots ≤0.1µm distant from nuclei. The length of each germ-line was defined using the Imaris Measurements tool. Measurement points were specifically placed at the beginning of the premeiotic tip and the end of pachytene. For germlines which had a defined transition zone by DAPI morphology, points were also placed at the start and end of the transition zone.

Nuclei positions were transformed from 3D coordinates to a linear order using the Gonad Linearization Algorithm implemented in R [32]. Gonad length in germlines which lacked a defined transition zone (e.g. *fog-2* mutants, S3 Fig) was normalized to the distance from the premeiotic tip to the end of pachytene, where the premeiotic tip begins at position 0 and the end of pachytene is at position 1. In all other germlines, the gonad length was normalized to pachytene, where the beginning of pachytene is position 0 and the end of pachytene is position 1. The accumulation of RAD-51 foci in *rad-54* mutant germlines (S1 Fig) was scored manually in 3D germlines using Imaris. Nuclei were rotated in 3D to ensure the identification of all RAD-51 foci. A RAD-51 focus was counted if the observed signal: 1) was associated with DAPI signal; and, 2) was brighter and larger than internuclear background staining, if any was present.

Germline DSB-2 staining was analyzed in Imaris using germlines stitched in Fiji as described above. The length of the germline was defined using the Imaris Measurements tool. Specific points were placed at the beginning of the transition zone, end of the transition zone, beginning of the DSB-2 zone (defined as the row of nuclei in which most nuclei had DSB-2 staining), the end of the DSB-2 zone, the final position of one or more nuclei which had DSB-2 staining, and the end of pachytene. The measured distances were then normalized to pachytene, where the beginning of pachytene is position 0 and the end of pachytene is position 1.

GFP::COSA-1 foci were quantified manually from late pachytene nuclei from 3D z-stacks using Fiji. Nuclei that were completely contained within the image stack were quantified in the last few rows of the late pachytene region in which all nuclei displayed bright GFP::COSA-1 foci (last ~3–6 rows of late pachytene in old *fog-2* germlines, last ~6–8 rows of late pachytene in young *fog-2* germlines).

### *fog-2* brood viability assay

*C. elegans* strains were maintained at 20˚C during fertility assays. Feminized *fog-2* mutants were synchronized in age by placing gravid mated CB4108 females onto an NGM plate seeded with OP50 for one hour. Hatched female progeny were isolated as L4s from these plates and were kept in isolation from males to prevent mating. At adult day 1, 2, 3, 4, or 5, these isolated *fog-2* females were then placed on individual plates with n = 2 young adult N2 males each. Mated *fog-2* females were then subsequently transferred to new NGM plates seeded with OP50 with young adult N2 males at either 6hr, 12hr, 18hr, 24hr, and 48hr after the first mating, or at 24hr and 48hr after the first mating. 72hr after the first mating, adult females were discarded. Plates were scored ~24hr after the parent female was removed for hatched progeny, dead eggs,

and unfertilized oocytes. Brood viability was calculated as (hatched progeny) / (hatched progeny + dead eggs). Fertility assays were replicated twice with n = 5 females of each age group assayed per replicate.

During the course of the brood viability assays, some mated *fog-2* females exhibited matricidal hatching. This phenotype was more pronounced in aged worms, consistent with previous work which showed that matricidal hatching is exacerbated with maternal age [33]. Only eggs which were successfully ovulated were scored in the assay.

### *uev-2* brood viability and incidence of males assay

*C. elegans* strains were maintained at 20˚C during fertility assays. n = 5 L4 wildtype and *uev-2* hermaphrodites were isolated and ~20 hours later were singled to individual NGM plates seeded with OP50. ~24 hours after singling, the parent hermaphrodites were discarded. ~24 hours after the parent hermaphrodites were removed, the plates were scored for unhatched eggs. ~48 hours after the hermaphrodites were removed, the hatched F1 progeny were scored as hermaphrodites or males. Brood viability was calculated as (hatched progeny) / (hatched progeny + dead eggs). Incidence of males was calculated as (male progeny) / (male progeny + hermaphrodite progeny).

### DAPI body quantification

L4 hermaphrodites were isolated on NGM plates seeded with OP50 ~20 hours before DAPI body quantification and were maintained at 20˚C. These worms were then picked to a Super-Frost Plus slide (VWR) in M9 buffer and were fixed 3x in 95% EtOH. 20μL of a solution containing 50% VECTASHIELD mounting media (Vector laboratories) and 50% 2μg/mL DAPI in ddH$_2$O was applied to each slide, and the slides were mounted with a No 1.5 coverslip (VWR) and sealed with nail polish. DAPI staining bodies were scored the same day as the slides were made on a Leica DM5500B microscope using a 60x objective. Only oocytes in the -1 to -3 positions were included in this analysis.

### Statistics

All statistics were calculated in R (v4.0.3). Data wrangling was performed using the Tidyverse package (v1.3.0) [34]. Specific statistical tests used are denoted in the Fig legends and text. P values were adjusted for multiple comparisons when appropriate. If 3 pairwise comparisons were being performed, Bonferroni correction was applied. If >3 pairwise comparisons were performed, Holm-Bonferroni correction was instead applied to reduce the risk of type II statistical errors.

### Data and code availability

All data are fully available without restriction. The spreadsheets delineating the raw numerical data used in this study are included in supporting information. These files include our COSA-1 counts in *fog-2* mutants, the brood viability counts in *fog-2* mutants, the brood viability and incidence of males counts in wild type and *uev-2* mutants, DAPI body counts in wild type and *uev-2* mutants, RAD-51 counts in *rad-54* mutants, and RAD-51 counts in wild type, *uev-2*, *fog-2*, and *pie-1p::uev-2* mutants. The gonad linearization algorithm is available on the Libuda Lab GitHub <github.com/libudalab/Gonad-Analysis-Pipeline> and on the Libuda Lab website <libudalab.org>.

## Results

### Meiotic DNA break levels are influenced by both aging and sperm depletion

To determine the relative contributions of reproductive aging and sperm depletion to DNA break repair dynamics in the *C. elegans* germline, we examined DNA break levels in the oocytes of aged hermaphrodites which were mated (to prevent sperm depletion) or unmated (to permit sperm depletion) (Fig 1A). DNA breaks were quantified using immunofluorescence to visualize the recombinase RAD-51, which marks DSBs designated for repair by recombination [35]. While the cytological appearance of RAD-51 foci indicates the occurrence of DSBs, disappearance of RAD-51 foci indicates progression of a DSB event further through a DSB repair pathway. "Young" germlines were isolated from N2 hermaphrodites on the first day of adulthood (1 day post-L4, Fig 1A), while "aged" germlines were isolated from N2 hermaphrodites on their fourth day of adulthood (4 days post-L4, Fig 1A). Aged hermaphrodites were maintained either unmated to males, or mated with males from their second to third day post-L4 larval stage (Fig 1A, see Methods).

To quantify the profile of DSB induction and repair across prophase I, we counted the number of RAD-51 foci per nucleus in oocytes from young and aged animals throughout the germline (see Methods). Under normal conditions, RAD-51 foci accumulate within nuclei following DSB induction by the conserved endonuclease SPO-11 in early pachytene [2,36]. Then, as nuclei progress through mid and late pachytene, these RAD-51 foci decline in number as DSBs are repaired [36]. During early pachytene, the amount of RAD-51 foci per nucleus was similar between aged mated germlines and young germlines (Fig 1B and 1C, Bin 2 young (RAD-51 foci per nucleus mean ± standard deviation) young 3.5±2.8, aged mated 3.4±3.8 Mann-Whitney U test p = 0.258). Young germlines, however, accumulated a slightly higher total number of RAD-51 foci per nucleus (Fig 1B and 1C, Bin 3 young 6.9±3.9, aged mated 6.5 ±5 Mann-Whitney U test p = 0.005), suggesting that DSB induction or RAD-51 loading is marginally compromised in aged mated germlines. We further noted that RAD-51 foci in aged unmated germlines were greatly decreased throughout early pachytene as compared to both young and aged mated germlines (Fig 1B and 1C, Bin 2 young 3.5±2.8, aged mated 3.4±3.8, aged unmated 1.4±1.5 Mann-Whitney U test p<0.001; Bin 3 young 6.9±3.9, aged mated 6.5±5, aged unmated 4±3.2 Mann-Whitney U test p<0.001). To test whether the reduced early pachytene RAD-51 foci observed in aged unmated germlines may be explained by increased turnover of RAD-51 at this stage, we also assessed the accumulation of DSBs via RAD-51 staining in *rad-54* mutants, which is defective for RAD-51 removal and progression of meiotic recombination [37,38]. We found that RAD-51 foci accumulation was delayed specifically in aged *rad-54* germlines and not aged mated *rad-54* germlines (S1 Fig). Taken together, the results of our experiments in aged wildtype and *rad-54* mutant germlines are concordant with a model in which sperm depletion specifically downregulates DSB induction and/or RAD-51 loading in early pachytene.

In contrast to early pachytene, nuclei throughout mid pachytene from aged mated germlines maintained higher levels of RAD-51 than young germlines (Fig 1B and 1C, Bin 4 young 5.0±4.1, aged mated 5.8±4.5 Mann-Whitney U test p = 0.003; Bin 5 young 2.4±2.0, aged mated 4.2±5.2, Mann-Whitney U test p = 0.017). We observed a similar effect in the aged unmated germlines, which also displayed elevated numbers of RAD-51 foci relative to young germlines throughout mid pachytene (Fig 1B and 1C, Bin 4 young 5.0±4.1, aged unmated 6.6±3.9 Mann-Whitney U test p<0.001; Bin 5 young 2.4±2.0, aged unmated 4.7±4.5 Mann-Whitney U test p = 0.001). Thus, DSB repair at mid-pachytene may be delayed in aging germlines regardless of mating or sperm depletion. Notably, by late pachytene the number of RAD-51 foci per

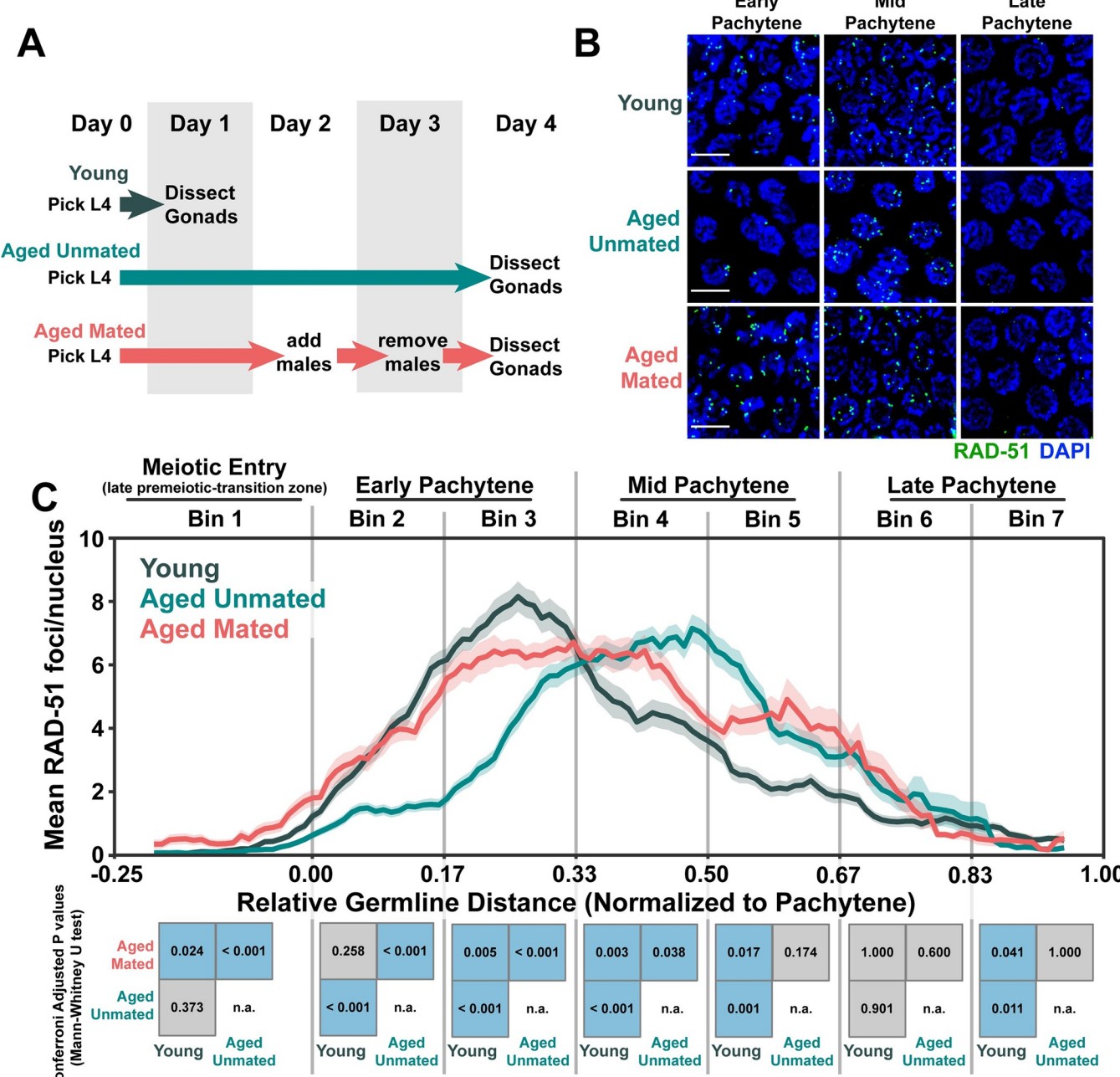

**Fig 1. DNA double strand break levels are altered during *C. elegans* germline aging.** A) Schemes used to isolate young (1 day post-L4) and aged (4 days post L4) worms for experiments. Days count ~18–24 hour periods after hermaphrodites were isolated as L4 larvae and are separated by alternating grey shaded boxes. B) Representative images of RAD-51 foci in meiotic nuclei of young and aged germlines. Scale bars represent 5μm. C) RAD-51 foci per nucleus in oocytes. Line plots represent the mean RAD-51 foci per nucleus along the length of the germline in a sliding window encompassing 0.1 units of normalized germline distance with a step size of 0.01 germline distance units. Mean RAD-51 foci were calculated from nuclei analyzed in n = 9 total germlines derived from ≥3 experimental replicates within each age group. Shaded areas around each line represent ± SEM. Total nuclei analyzed (Bins 1/2/3/4/5/6/7) Young: 185/117/146/107/117/97/83; Aged Mated: 234/205/192/173/154/177/96; Aged Unmated: 268/147/161/175/191/152/110. Germline distances were normalized to the start (0) and end (1) of pachytene based on DAPI morphology (see Methods). For analysis, the germline was divided into 7 bins encompassing the transition zone (Bin 1), early pachytene (Bins 2–3), mid pachytene (Bins 4–5), and late pachytene (Bins 6–7). The germline positions at which each bin start and end are marked on the X axis as vertical grey lines. Heat maps below each bin display the p values of pairwise comparisons of RAD-51 foci per nucleus counts within that bin. P values were calculated using Mann-Whitney U tests with Bonferroni correction for multiple comparisons. Blue tiles indicate significant differences (adjusted p value <0.05) and grey tiles indicate nonsignificant effects (adjusted p value >0.05). n.a. = not applicable. Numerical data associated with this figure is presented in S1 Data.

nucleus converged between young, aged mated, and aged unmated germlines (Fig 1B and 1C, Bin 6 young 1.6±2.2, aged mated 2.0±4.9, aged unmated 1.9±4.3 Mann-Whitney U test p>0.05), indicating that ultimately all DSBs can be repaired or minimally offload RAD-51 in aged germlines. Taken together, our results suggest that parallel mechanisms may regulate DNA break levels in aged *C. elegans* germlines: 1) depletion of sperm downregulates DSB induction and/or RAD-51 loading; and, 2) reproductive aging delays RAD-51 foci unloading at mid pachytene.

To determine if the persistent RAD-51 foci in aged mated and unmated germlines were derived from the programmed meiotic DSBs, we also examined RAD-51 foci in *spo-11(me44)* null mutants, which do not form meiotic DSBs (S2 Fig) [36]. We did not observe a notable increase in nuclei with RAD-51 foci in aged *spo-11* germlines, indicating that the persistent RAD-51 foci present at mid pachytene in aged wildtype gonads are likely derived from normal meiotic functions and processes, such as SPO-11 activity.

Nuclei which are competent for DSB induction in the *C. elegans* germline have their chromatin marked with the protein DSB-2 [30]. To assess if the altered accumulation of DSBs which we observed in aged unmated germlines coincided with a change in competency for DSB induction, we quantified the extent of young and aged germlines in which ≥50% of nuclei exhibited DSB-2 staining (the "DSB-2 zone", S3A and S3B Fig). DSB-2 accumulates on meiotic chromatin beginning in the transition zone (leptotene/zygotene) and is offloaded from the majority of nuclei by mid pachytene [30,32]. Mutants which incur errors in crossover formation, however, maintain DSB-2 on meiotic chromatin later into pachytene [30]. While the length of the DSB-2 zone was only subtly altered in aged mated germlines (S3B and S3C Fig, Mann-Whitney U test p = 0.027), the DSB-2 zone persisted later into pachytene in aged unmated germlines relative to young germlines (S3B and S3C Fig, Mann-Whitney U test p = 0.008). Thus, our data indicate that the extent of DSB-2 marked pachytene nuclei is influenced by the depletion of sperm as well as by aging, albeit to a smaller extent.

## Meiotic DNA breaks are elevated in aged feminized germlines

To uncouple the relationship between sperm depletion and reproductive aging in regulating DSB induction and repair, we examined RAD-51 levels in germlines which have never been impacted by sperm or mating. Hermaphrodites carrying the *fog-2(q71)* mutation do not produce sperm during larval development but proliferate a full adult complement of oocytes [16], rendering them "obligate females." Due to the absence of signaling from sperm in *fog-2* mutants, both germline stem cell proliferation and meiotic progression are halted, such that meiotic oocytes are held within the gonad [14,15,18]. Nonetheless, feminized mutants undergo reproductive senescence and exhibit reduced oocyte quality with age (S4 Fig; [5,12])

We analyzed the levels of RAD-51 foci in oocyte nuclei from young (1 day post-L4), aged (4 days post-L4), and old (6 days post-L4) *fog-2* germlines (Fig 2A and 2B). During our experiments we noted that the cytologically distinctive transition zone, which demarcates meiotic entry and is composed of nuclei undergoing active chromosome movement to facilitate pairing, was dramatically reduced in aged *fog-2* germlines (S5 Fig). Previous work has shown that mitotic germ cell proliferation is reduced in feminized and sperm-depleted germlines [14]. Thus, the absence of a transition zone in aged *fog-2* germlines may be the product of two parallel effects: 1) nuclei in the transition zone completing the pairing process and therefore exhibiting the classic "cage-like" morphology of paired chromosomes found in pachytene nuclei; and, 2) decreased proliferation also limiting the number of new nuclei which enter meiosis. This lack of the transition zone in aged *fog-2* germlines presented a challenge for staging meiotic nuclei to make comparisons between young and aged gonads. To quantify RAD-51 levels in

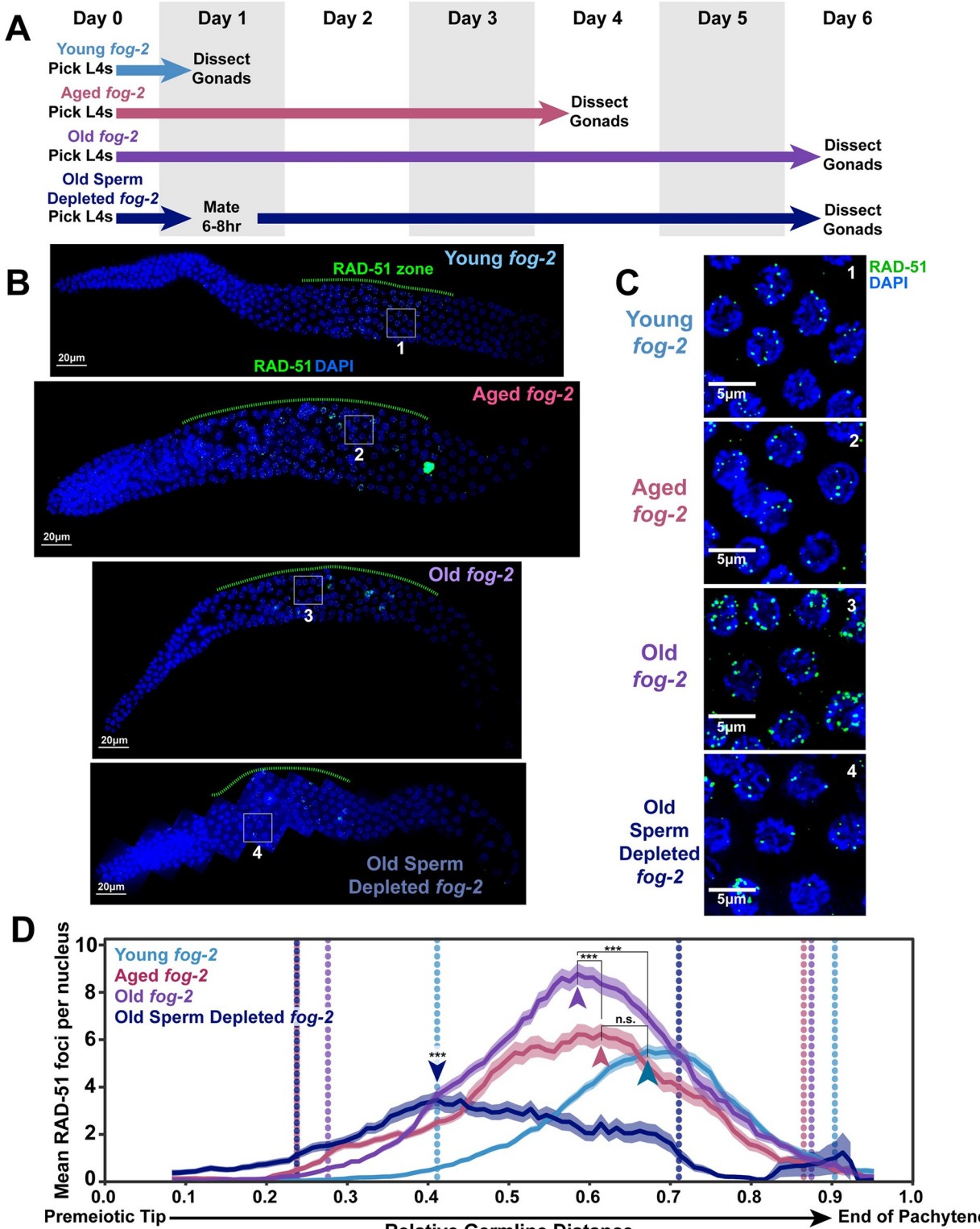

**Fig 2. DNA double strand break levels increase with age in *fog-2* feminized germlines.** A) Schemes used to isolate young (1 day post-L4), aged (4 days post L4), and old (6 days post L4) *fog-2(q71)* worms for experiments. Days count ~18–24 hour periods after hermaphrodites were isolated as L4 larvae and are separated by alternating grey shaded boxes. B) Representative whole germline images of young, aged, and old *fog-2* germlines. The RAD-51 zone is indicated with a green dashed line. All germlines are oriented with the distal mitotic tip on the left and the end of pachytene on the right. Scale bars represent 20μm. Grey numbered boxes indicate the positions of the images presented in panel C. C)

Representative images of the peak levels of RAD-51 foci observed in meiotic nuclei of young, aged, and old *fog-2* germlines. Scale bars represent 5μm. Each panel is numbered to indicate the position in the germlines displayed in panel B that each inset was taken from. D) RAD-51 foci per nucleus in *fog-2(q71)* oocytes. Line plots represents the mean RAD-51 foci per nucleus along the length of the germline in a sliding window encompassing 0.1 units of normalized germline distance with a step size of 0.01 germline distance units. Mean RAD-51 foci were calculated from nuclei analyzed in n = 9 total germlines for young, aged, and old *fog-2* and from n = 7 total germlines for aged sperm depleted *fog-2*. For all groups, the analyzed germlines were derived from ≥3 experimental replicates. Shaded areas around each line represent ± SEM. Average nuclei quantified in each bin ± standard deviation: Young *fog-2* 173±21.3, Aged *fog-2* 143±32.0, Old *fog-2* 148.5±29.6, Aged Sperm Depleted *fog-2* 119 ±40.5. Germline distance was normalized to the premeiotic tip (0) and end of pachytene (1) based on DAPI morphology (see Methods). Arrowheads indicate the "peak RAD-51" windows, defined as the windows along the length of the germline of each age group with the highest RAD-51 foci per nucleus. P values were calculated by Mann-Whitney U test comparisons of RAD-51 counts within these peak windows with Holm-Bonferroni correction for multiple comparisons (n.s. = p>0.05, *** = p<0.001). Old sperm depleted germlines exhibited significantly less RAD-51 foci than all other groups within the peak RAD-51 window. Vertical dotted lines indicate the distal and proximal bounds of the RAD-51 zone for each age group. Numerical data associated with this figure is presented in S1 Data.

*fog-2* germlines independent of meiotic stages, we normalized the germline length with position 0 at the premeiotic tip and position 1 at the end of pachytene and used a sliding window to assay RAD-51 foci within the germline (Fig 2D, see Methods) [32]. To describe the RAD-51 profile of aging *fog-2* germlines, we calculated two metrics: 1) the "RAD-51 zone" indicating the extent of the germline which contained nuclei with RAD-51 foci; and, 2) the "peak RAD-51 window" indicating the maximum levels of RAD-51 within the germlines.

To assess whether the proportion of germline nuclei with RAD-51 foci was altered in aging *fog-2* germlines, we calculated the "RAD-51 zone" of each age group, which was defined as the germline distance extending from the most distal (near the premeiotic tip) to the most proximal (near the end of pachytene) windows in which at least 50% of nuclei had one or more RAD-51 foci (Fig 2B and 2D). We found that the RAD-51 zone extended more distally in the germline in aged and old *fog-2* animals as compared to young germlines (Fig 2B and 2D). This distal expansion of the RAD-51 zone can likely be explained by transition zone nuclei in young germlines completing the pairing process and entering pachytene as the germline ages. In contrast, the proximal end of the RAD-51 zone only subtly shifted distally in aged and old germlines (Fig 2B and 2D). This result indicates that later prophase I nuclei within aged *fog-2* germlines continue to either maintain or induce RAD-51 marked DSBs.

To determine if the number of RAD-51 marked DSBs in *fog-2* germline nuclei were altered with age, we identified the "peak RAD-51 windows" in each age group, defined as the window in which the mean RAD-51 foci per nucleus was highest (Fig 2C and 2D arrowheads). We noted that the position of the peak RAD-51 window moved distally in aged and old *fog-2* germlines (Fig 2D arrowheads), suggesting that the spatial regulation of DSB induction and repair may change as feminized germlines age. The number of RAD-51 foci per nucleus within the peak RAD-51 window was not significantly different in aged germlines as compared to young gonads (Fig 2C and 2D, young *fog-2* 5.5±3.6, aged *fog-2* 6.2±5.3 Mann-Whitney U test p = 1.000). Old germlines, however, exhibited a significant ~1.5-fold increase in RAD-51 foci per nucleus as compared to young and aged germlines within the peak RAD-51 window, indicating that *fog-2* mutant germ cells accumulate RAD-51 foci during aging (Fig 2C and 2D, young *fog-2* 5.5±3.6, aged *fog-2* 6.2±5.3, old *fog-2* 8.8±5.2 Mann-Whitney U test p <0.001). This result notably differs from aged unmated wildtype germlines, which exhibit reduced DSBs with age (Fig 1B and 1C) [8,10]. Our data therefore support a model in which sperm depletion, rather than absence of sperm, downregulates meiotic DSB induction.

To test whether sperm depletion is sufficient to downregulate the levels of RAD-51 marked DSBs in meiotic nuclei, we mated *fog-2* females 1 day post-L4 and then allowed the females to deplete this sperm supply over five subsequent days (Fig 2A). We observed that the peak RAD-51 levels were dramatically reduced in these old sperm-depleted germlines relative to young, aged, and old unmated *fog-2* mutants (Fig 2C and 2D, young *fog-2* 5.5±3.6, aged *fog-2* 6.2±5.3,

old *fog-2* 8.8±5.2, old sperm depleted *fog-2* 3.5±0.4 Mann-Whitney U test p <0.001). Additionally, we noted that the RAD-51 window was shortened in aged sperm depleted *fog-2* germlines (Fig 2B and 2D), demonstrating that a reduced proportion of nuclei in these germlines maintained DSBs as compared to their unmated counterparts. Thus, our data strongly indicate that the reduced accumulation of RAD-51 marked DSBs observed during the normal course of *C. elegans* aging is primarily regulated by sperm depletion, rather than aging *per se*.

## DSB repair is altered in aged feminized germlines

The accumulation of RAD-51 foci observed in aging unmated *fog-2* germlines may be the product of: 1) increased induction of DSBs; 2) defects in DSB repair; or, 3) a combination of these effects. To assess the efficiency of DSB repair during *fog-2* germline aging, we exposed young (1 day post-L4) and aged (4 days post-L4) *fog-2* mutant females to 5000 Rads of ionizing radiation (Figs 3A and S6), inducing ~118 DSBs per nucleus throughout the germline [31]. We then allowed the animals to age for 2 days to resolve this DNA damage before assessing germlines for persistent unrepaired DSBs as marked by RAD-51 foci (Fig 3A). As *fog-2* germlines accumulate DSBs during aging (Fig 2C and 2D), we established baseline levels of DNA damage based on comparing RAD-51 foci in animals of equivalent ages that were never exposed to radiation to the irradiated cohorts (Fig 3A). We noted considerable inter-nucleus variance in the RAD-51 foci, which persisted following irradiation in both young and aged germlines (S6 Fig). This effect was particularly prominent in the distal germlines of both groups (S6C and S6D Fig), suggesting that a subpopulation of nuclei in the mitotic germline or early stages of meiosis are uniquely susceptible to exogenous DNA damage regardless of parental age.

To estimate the residual DSBs derived from irradiation which were not yet repaired two days post irradiation, we calculated the median number of RAD-51 foci in a sliding window across the germline (Fig 3B, see Methods) and subtracted the unirradiated median RAD-51 foci from the irradiated median RAD-51 foci in each window (Fig 3C). The median RAD-51 foci was selected to limit the potential of high damage outlier nuclei to disproportionately impact our analysis (S6 Fig). Both young and aged germlines maintain high levels of damage in the distal germline following irradiation (germline position 0.0–0.5, Figs 3C and S6C). Nuclei in the proximal region of young irradiated germlines did not consistently maintain median DNA break levels higher than baseline; whereas aged irradiated germlines maintained a median elevation of ~6–10 RAD-51 foci per nucleus (germline position 0.5–1.0, Fig 3C). This result indicates that aged *fog-2* germlines exhibit DNA repair defects specifically in nuclei in the proximal germline, suggesting that DSB repair defects are incurred at later stages of meiotic prophase I during aging. Our result parallels previous experiments in *spo-11* mutants, which revealed that repair of irradiation-induced DSBs is delayed in aged germlines which have been exposed to sperm [8]. Taken together, our experiments in "feminized" mutants demonstrate that DNA repair efficiency is altered in aging germlines independent of any signals from sperm.

Defects in meiotic DSB repair may disrupt the formation of interhomolog crossovers [35], and previous work has shown that aged unmated wildtype germlines exhibit reduced COSA-1 foci and bivalent formation [8,10]. To determine if crossover recombination is impeded in aged *fog-2* germlines, we quantified bright GFP::COSA-1 foci, which mark joint molecules designated to become crossovers at late pachytene [31]. Under normal conditions, each of the six pairs of *C. elegans* homologous chromosomes form one crossover and therefore most nuclei exhibit only six COSA-1 foci [31]. We observed that the number of COSA-1 foci in late pachytene nuclei of old unmated *fog-2* gonads were altered relative to their young counterparts (Fig

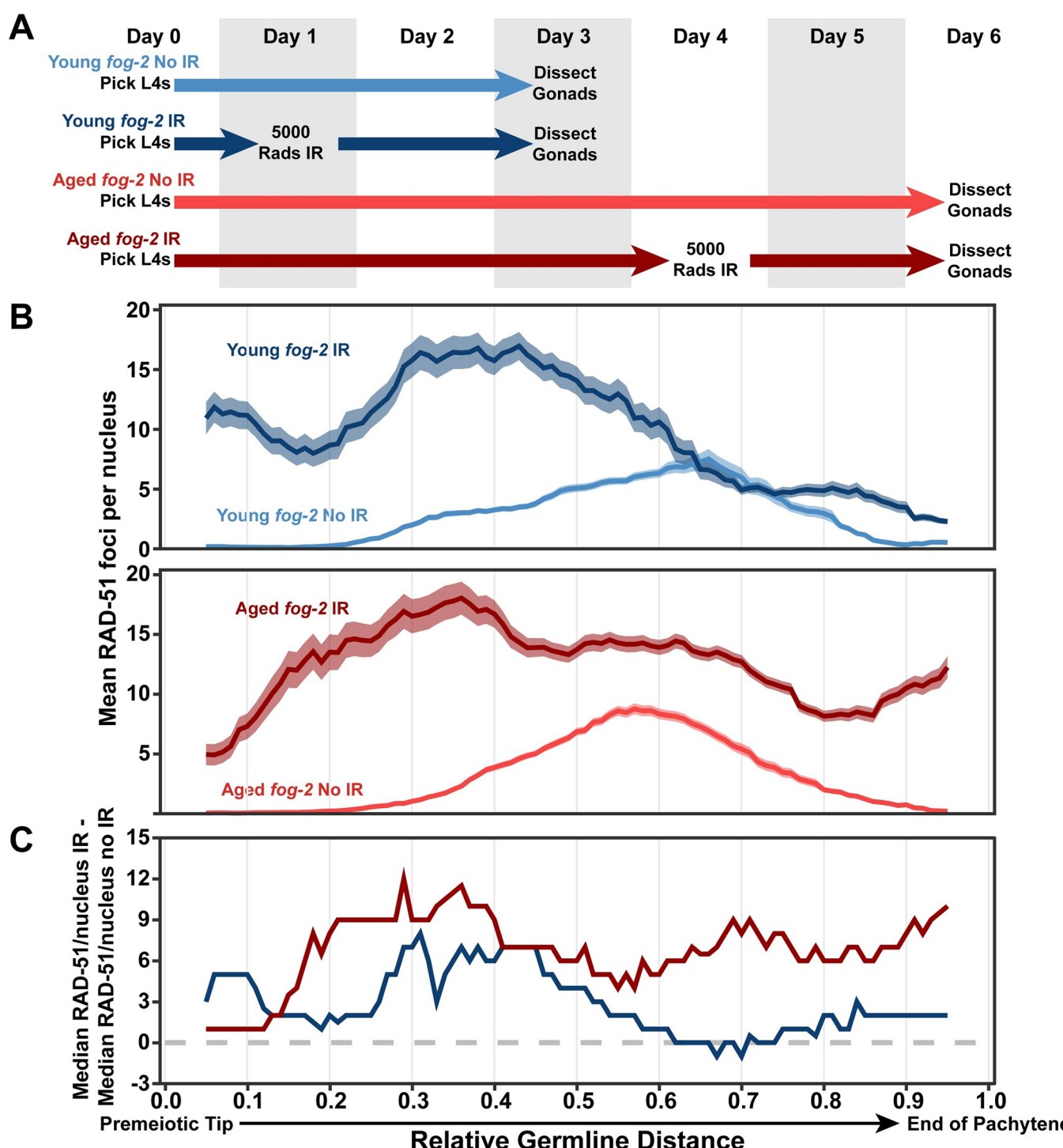

**Fig 3. DNA double strand break repair is disrupted in aged *fog-2* feminized germlines.** A) Schemes used to isolate young and aged *fog-2(q71)* worms for experiments. Note that 'young' worms are 3 days post-L4 and 'aged' worms are 6 days post-L4 at the time of dissection in this Fig. Days count ~18–24 hour periods after hermaphrodites were isolated as L4 larvae and are separated by alternating grey shaded boxes. Irradiated (IR) germlines were exposed to ionizing radiation at the first or fourth day of adulthood and were allowed to recover for two days before analysis (see Methods). Unirradiated germlines (No IR) were never exposed to radiation. B) RAD-51 foci per nucleus in irradiated (IR) and unirradiated (no IR) oocytes. Line plots represents the mean RAD-51 foci per nucleus along the length of the germline in a sliding window encompassing 0.1 units of normalized germline distance with a step size of 0.01 germline distance units. Plots in panel B share an X axis with the plot in panel C. Mean RAD-51 foci were calculated from nuclei analyzed in n = 9 total germlines derived from ≥3 experimental replicates within each age group. Shaded areas around each line represent ± SEM. Average nuclei quantified in each bin ± standard deviation: Young *fog-2* No IR 141.6±26.6, Young *fog-2* IR 134.4±27.9, Old *fog-2* No IR 148.5±29.6, Old *fog-2* IR 142.4±30. Germline distance was normalized to the

premeiotic tip (0) and end of pachytene (1) based on DAPI morphology (see Methods). Representative images of young and aged IR and No IR germlines are displayed in S5 Fig. C) Median RAD-51 foci per nucleus in irradiated germlines above median levels in unirradiated germlines of the same age (calculated as median RAD-51 foci in IR gonads–median RAD-51 foci in non-IR gonads within each window along the length of the germline). Old *fog-2* unirradiated RAD-51 foci data is shared with Fig 2. Numerical data associated with this figure is presented in S1 Data.

4A, Chi square test p = 0.033), with a subtle but significant increase specifically in the fraction of nuclei with >6 COSA-1 foci (Fig 4A, Fisher's Exact test p = 0.038). A similar effect was observed for old sperm depleted *fog-2* gonads, which also exhibited altered distributions of COSA-1 foci in late pachytene (Fig 4A, Chi square test p = 0.037), and an increase in nuclei with >6 COSA-1 foci (Fig 4A, Fisher's Exact test p = 0.029). Thus, our data suggests that DSB repair in old *fog-2* germlines may be altered in a manner which impacts the number of designated crossovers, and that this change is independent of sperm depletion.

Further, we noted that the late pachytene region of the germline with bright COSA-1 foci was reduced in length in old *fog-2* germlines but was not reduced in old *fog-2* sperm depleted germlines (Fig 4B). *fog-2* mutants accumulate diakinesis oocytes over time [39], so we can infer that a proportion of nuclei with bright COSA-1 foci in young germlines may progress into diplotene and diakinesis as the worm ages even in the absence of sperm. The reduction in COSA-1 marked nuclei in old unmated *fog-2* germlines therefore suggests that in the absence

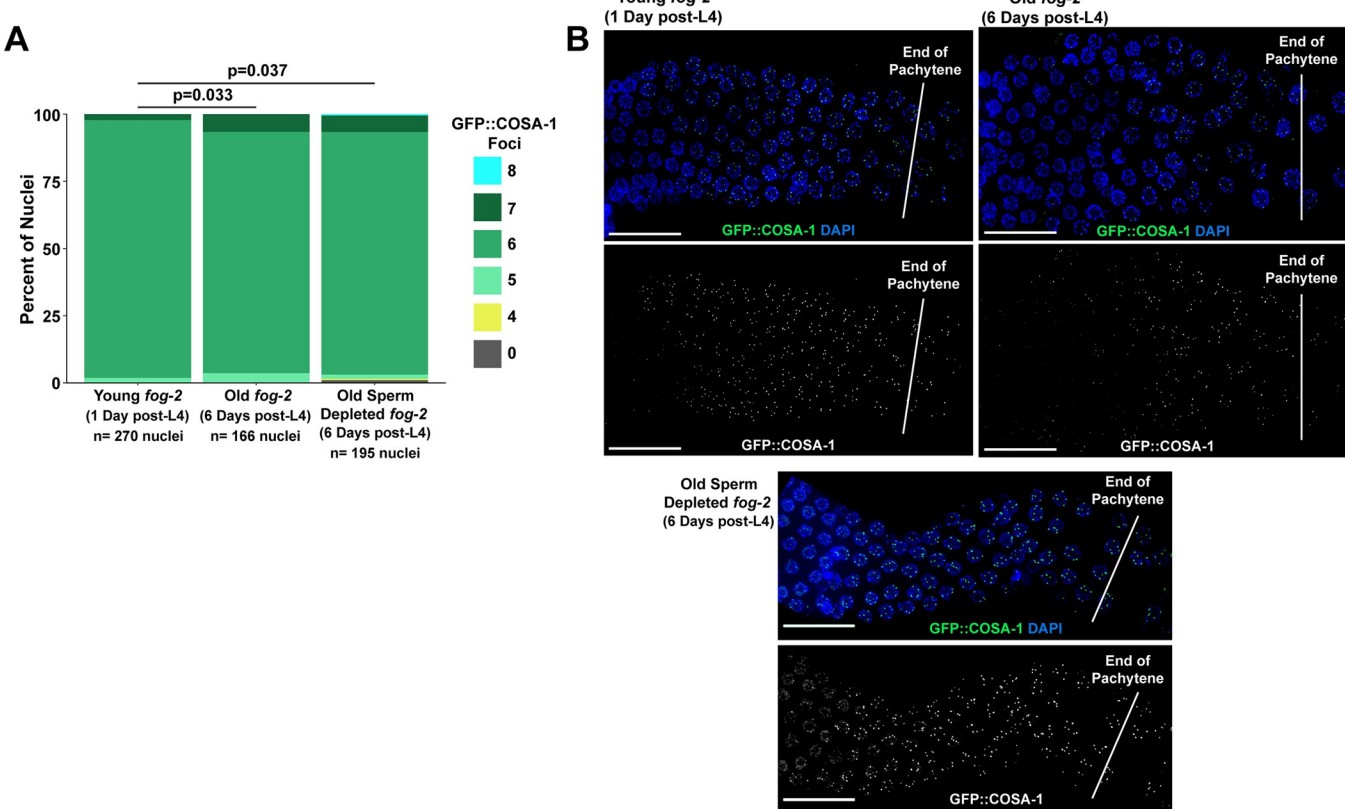

**Fig 4. Crossover designation is altered in aged *fog-2* germlines.** A) Stacked bar plot of the percent of nuclei in young (1 day post L4) and old (6 days post L4) unmated or sperm depleted *fog-2* mutants with the given number of GFP::COSA-1 foci. See Fig 2A for an outline of the maintenance schemes used in each of these cohorts. The p value displayed was calculated by Chi Square test comparing the proportion of nuclei with >6, 6, or <6 COSA-1 foci. N values indicate the number of nuclei scored. B) Representative images of GFP::COSA-1 localization in young and old *fog-2* mutant germlines. Scale bars represent 20μm. Germlines are oriented with the proximal end on the left and distal end on the right. Vertical low opacity white lines designate the end of pachytene. Numerical data associated with this figure is presented in S4 Data.

of sperm, mid-pachytene nuclei may not progress through subsequent meiotic stages, including crossover designation, at a robust rate. Previous work has shown that aged feminized *C. elegans* mutants exhibit a delay in optimal progeny production and temporarily deplete their accumulated supplies of nuclei in diplotene and diakinesis after mating [14]. Our data raises the possibility that the observed delay in progeny production in aged feminized mutants may be underpinned by the temporal requirements of resuming crossover designation and resolution preceding meiotic chromosome compaction.

## UEV-2 is required for 'youthful' germline DSB repair

To identify proteins which may regulate DSB repair in the aging *C. elegans* germline, we looked to candidate genes upregulated in long-reproductive *sma-2* mutant oocytes, which exhibit DNA damage resilience in addition to delayed reproductive senescence [4]. The *sma-2* DNA damage resilience phenotype requires upregulation of the E2 ubiquitin-conjugating enzyme variant UEV-2, suggesting that this protein may promote efficient germline DNA repair [4]. UEV proteins lack a catalytic cysteine residue conserved in E2 enzymes [40] but have been shown to form heterodimeric complexes with other E2 enzymes to influence their function, implying regulatory roles for this protein class [41,42].

To assess the influence of UEV-2 on DSB repair during germline aging, we utilized a strain carrying the putative null allele *uev-2(gk960600)*, which ablates the translation initiation site and second exon boundary of the gene (S7 Fig; see Methods). We determined that *uev-2* is not required for successful meiosis, as *uev-2* mutants exhibited normal progeny viability (wildtype 99.5% viable progeny n = 954, *uev-2* 99.3% viable progeny n = 914, Chi Square test p = 0.690), no evidence of elevated X chromosome nondisjunction (wildtype 0.1% male progeny n = 950, *uev-2* 0.1% male progeny n = 909, Chi Square test p = 0.975), and normal chiasma formation (diakinesis nuclei with 6 DAPI bodies wild type 100% n = 55, *uev-2* 100% n = 60, Fisher's exact test p = 1.000). With the *uev-2* mutant strain, we examined the number of RAD-51 foci in germline nuclei derived from young (1 day post-L4) or aged (4 days post-L4) animals (Fig 5A). Aged *uev-2* mutants were also mated to avoid the DSB induction defects associated with sperm depletion (Fig 5A). If UEV-2 functions to promote efficient DSB repair in young gonads but becomes dysregulated or loses function during aging, then we would expect *uev-2* mutants to exhibit defects in DSB repair in young germlines but minimal additional defects in aged germlines. Indeed, when we compared the levels of RAD-51 observed in young and aged mated wildtype and *uev-2* germlines, we observed DSB repair defects that did not accumulate with age. In early pachytene, young and aged *uev-2* mutants exhibited similar levels of RAD-51 to young wildtype germlines (Fig 5B and 5C, Bin 2 young wildtype 3.5±2.8, young *uev-2* 3.0 ±2.6, aged *uev-2* 4.0±3.2 Mann-Whitney U test p>0.05; Bin 3 young wildtype 6.9±3.9, young *uev-2* 6.6±4.1, aged *uev-2* 7.0±4.7 Mann-Whitney U test p>0.05), indicating that UEV-2 is not required for meiotic DSB induction nor RAD-51 loading. In contrast, at mid pachytene, young *uev-2* mutant germlines maintained elevated RAD-51 foci relative to young wildtype germlines (Fig 5B and 5C, Bin 4 young wildtype 5.0±4.1, young *uev-2* 5.3±3.3, aged *uev-2* 6.3 ±5.9 Mann-Whitney U test p<0.05; Bin 5 young wildtype 2.4±2.0, young *uev-2* 3.8±4.0, aged *uev-2* 5.0±7.3 Mann-Whitney U test p<0.05). The specific levels of DSBs at mid pachytene in young *uev-2* mutants were also indistinguishable from aged wildtype germlines (Fig 5B and 5C, Bin 4 young *uev-2* 5.3±3.3, aged wildtype 5.8±4.5 Mann-Whitney U test p>0.05; Bin 5 young *uev-2* 3.8±4.0, aged wildtype 4.2±5.2 Mann-Whitney U test p>0.05). These results at mid pachytene indicate that DSB repair is delayed in young *uev-2* mutants to an extent which recapitulates the effect we observe during wildtype aging. Aged *uev-2* germline RAD-51 levels at mid pachytene were statistically indistinguishable from either young or aged mated wildtype

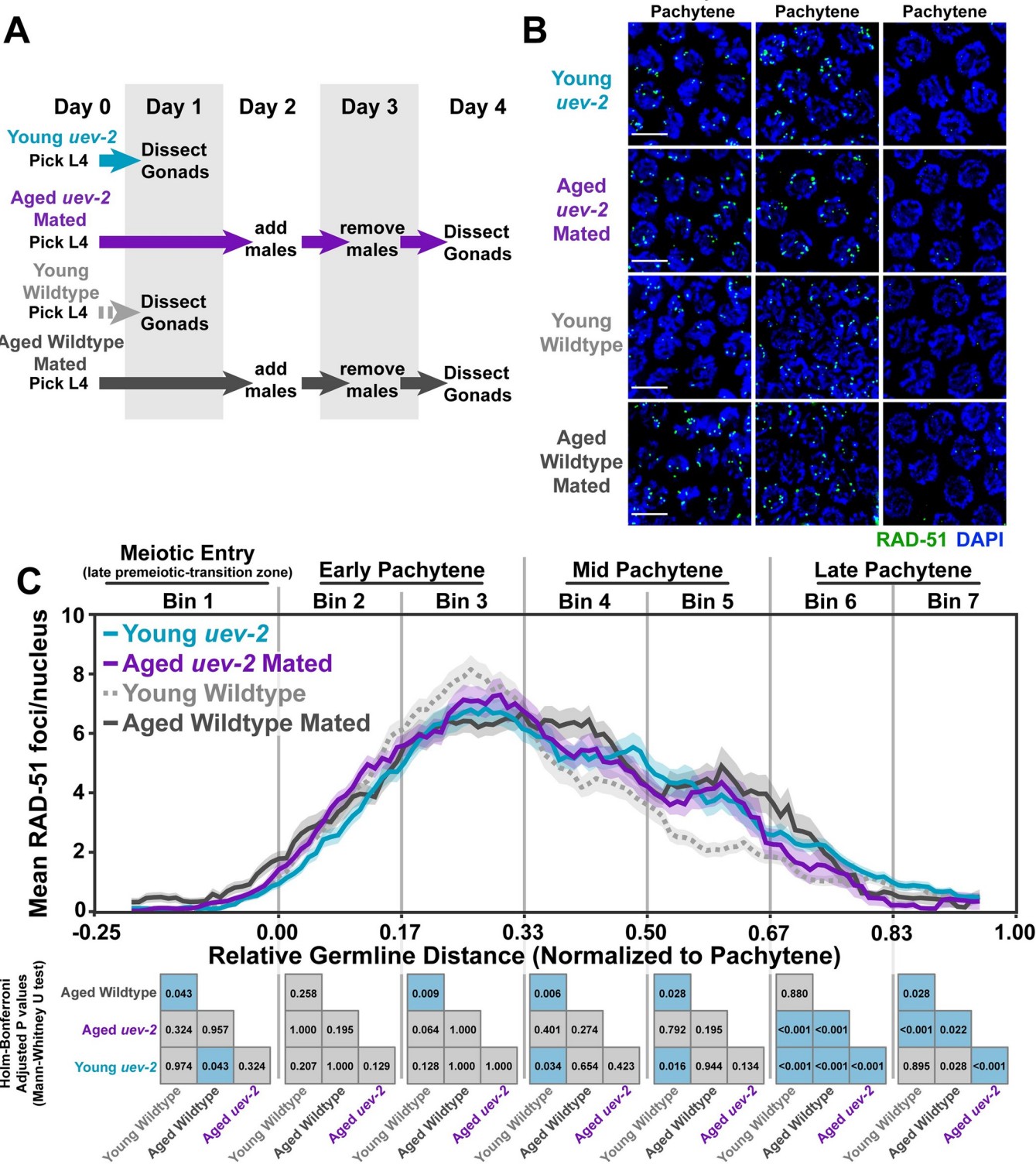

**Fig 5. UEV-2 is required for 'youthful' DNA repair.** A) Schemes used to isolate young (1 day post-L4) and aged (4 days post L4) *uev-2* mutant worms for experiments. Days count ~18–24 hour periods after hermaphrodites were isolated as L4 larvae and are separated by alternating grey shaded boxes. B) Representative images of RAD-51 foci in meiotic nuclei of young *uev-2*, aged mated *uev-2*, young wildtype, and aged mated wildtype germlines. Scale bars represent 5μm. C) RAD-51 foci per nucleus in oocytes. Line plots represent the mean RAD-51 foci per nucleus along the length of the germline in a sliding window encompassing 0.1 units of normalized germline distance with a step size of 0.01 germline distance units. Mean RAD-51 foci were calculated from

nuclei analyzed in n = 9 total germlines derived from ≥3 experimental replicates within each age and genotype group. Shaded areas around each line represent ± SEM. Total nuclei analyzed (Bins 1/2/3/4/5/6/7) Young Wildtype: 185/117/146/107/117/97/83; Aged Wildtype Mated: 234/205/192/173/154/177/96; Young *uev-2*: 186/135/134/113/112/132/95; Aged *uev-2* Mated: 129/161/155/158/167/142/120. Germlines distances were normalized to the start (0) and end (1) of pachytene based on DAPI morphology (see Methods). For analysis, the germline was divided into 7 bins encompassing the transition zone (Bin 1), early pachytene (Bins 2–3), mid pachytene (Bins 4–5), and late pachytene (Bins 6–7). The germline positions at which each bin start and end are marked on the X axis as vertical grey lines. Heat maps below each bin display the p values of pairwise comparisons of RAD-51 foci per nucleus counts within that bin. P values were calculated using Mann-Whitney U tests with Holm-Bonferroni correction for multiple comparisons. Blue tiles indicate significant differences (adjusted p value <0.05) and grey tiles indicate nonsignificant effects (adjusted p value >0.05). Young and aged mated wild type data is shared with Figs 1 and 6. Numerical data associated with this figure is presented in S1 Data.

germlines (Fig 5B and 5C, Bin 4 young wildtype 5.0±4.1, aged wildtype 5.8±4.5, aged *uev-2* 6.3 ±5.9 Mann-Whitney U test p>0.05; Bin 5 young wildtype 2.4±2.0, aged wildtype 4.2±5.2, aged *uev-2* 5.0±7.3 Mann-Whitney U test p>0.05), suggesting that the *uev-2* mutation does not grossly exacerbate DSB repair defects with age.

In late pachytene, the specific rates of DSB resolution diverged slightly between young and aged *uev-2* and wildtype germlines (Fig 5B and 5C, Bin 6 young wildtype 1.6±2.2, aged wildtype 1.9±4.3, young *uev-2* 1.9±1.9, aged *uev-2* 2.3±6 Mann-Whitney U test p<0.05; Bin 7 young wildtype 0.9±1.4, aged wildtype 0.5±1.9, young *uev-2* 0.7±1.0, aged *uev-2* 0.7±3.2 Mann-Whitney U test p<0.05), suggesting that UEV-2-independent and age-specific effects may contribute to DSB resolution at this meiotic stage. Taken together, our results indicate that loss of *uev-2* in young germlines is sufficient to phenocopy the mid-pachytene patterns of DSB repair observed in an aged wildtype context. This observation supports a model in which UEV-2 functions in young germlines specifically to promote efficient DSB repair.

## Overexpression of *uev-2* alters RAD-51 foci levels in aged oocytes

As loss of *uev-2* in young germlines appeared to "prematurely age" RAD-51 foci patterns, we hypothesized that overexpression of *uev-2* in aged germlines could ameliorate persistent RAD-51 foci at mid pachytene. To test this hypothesis, we used CRISPR/Cas9 genome editing to generate a germline-specific overexpression construct of *uev-2* driven by the *pie-1* promoter (*pie-1p::uev-2*, see Methods). We then assessed for the presence of DSBs as marked by RAD-51 in the germlines of young (1 day post-L4) or aged (4 days post-L4) mated animals overexpressing UEV-2 and compared those levels to young and aged mated wildtype germlines (Fig 6A).

At the beginning of early pachytene, both young and aged mated *pie-1p::uev-2* mutants initially accumulated DSBs at levels similar to young and aged mated wildtype gonads (Fig 6B and 6C, Bin 2 young wildtype 3.5±2.8, aged wildtype 3.4±3.8, young *pie-1p::uev-2* 3.4±2.7, aged *pie-1p::uev-2* 3.1±2.8 Mann-Whitney U test p>0.05). However, aged mated *pie-1p::uev-2* mutants accumulated fewer total DSBs than young wildtype, aged mated wildtype, and young *pie-1p::uev-2* germlines (Fig 6B and 6C, Bin 3 young wildtype 6.9±3.9, aged wildtype 6.5±5.0, young *pie-1p::uev-2* 6.8±3.3, aged *pie-1p::uev-2* 4.9±4.0 Mann-Whitney U test p<0.05). At mid pachytene, young *pie-1p::uev-2* germlines maintained elevated RAD-51 foci over young wildtype germlines, suggesting that overexpression of *uev-2* deleteriously impacted DSB repair in this context (Fig 6B and 6C, Bin 4 young wildtype 5.0±4.1, young *pie-1p::uev-2* 6.6±4.2 Mann Whitney U test p<0.001; Bin 5 young wildtype 2.4±2.0, young *pie-1p::uev-2* 4.5±3.7 Mann Whitney U test p<0.001). This effect was not preserved in aged mated *pie-1p::uev-2* germlines, which exhibited similar DSB levels as young wildtype germlines at the beginning of mid pachytene and slightly elevated foci at the end of mid pachytene (Fig 6B and 6C, Bin 4 young wildtype 5.0±4.1, aged *pie-1p::uev-2* 4.6±5.2 Mann-Whitney U test p = 0.156; Bin 5 young wildtype 2.4±2.0, aged *pie-1p::uev-2* 2.9±5.3 Mann-Whitney U test p = 0.023).

Throughout late pachytene, young *pie-1p::uev-2* germlines maintained subtle but significantly elevated DSBs relative to young and aged wildtype germlines (Fig 6B and 6C, Bin 6

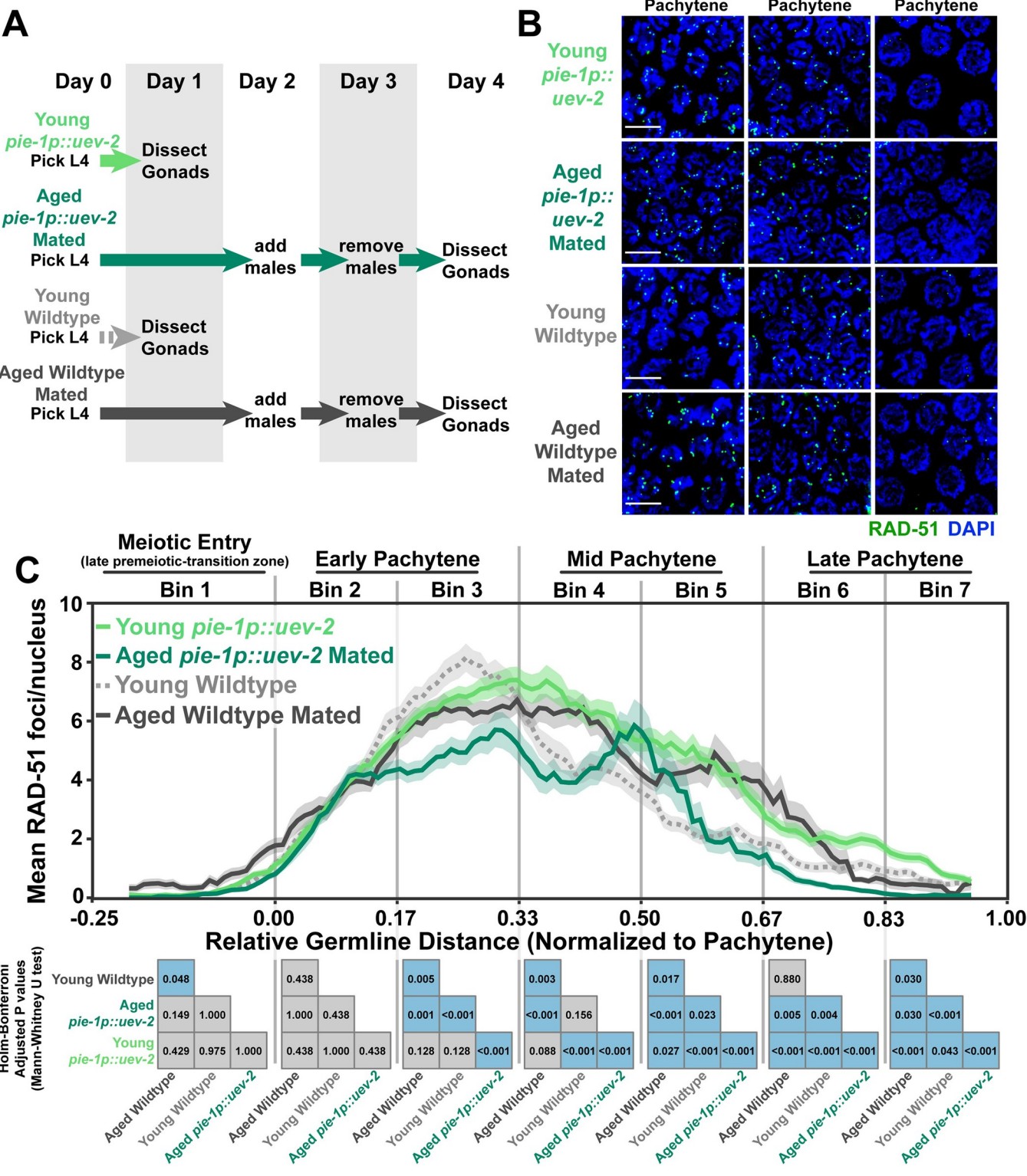

**Fig 6. Germline *uev-2* overexpression differentially impacts DSB levels in young and aged germlines.** A) Schemes used to isolate young (1 day post-L4) and aged (4 days post L4) worms for experiments. Days count ~18–24 hour periods after hermaphrodites were isolated as L4 larvae and are separated by alternating grey shaded boxes. B) Representative images of RAD-51 foci in meiotic nuclei of young *pie-1p::uev-2*, aged *pie-1::uev-2*, and young wildtype germlines. Scale bars represent 5μm. C) RAD-51 foci per nucleus in oocytes. Line plots represent the mean RAD-51 foci per nucleus along the length of the germline in a sliding

window encompassing 0.1 units of normalized germline distance with a step size of 0.01 germline distance units. Mean RAD-51 foci were calculated from nuclei analyzed in n = 9 total germlines derived from ≥3 experimental replicates within each age group. Shaded areas around each line represent ± SEM. Total nuclei analyzed (Bins 1/2/3/4/5/6/7) Young Wildtype: 185/117/146/107/117/97/83; Aged Wildtype Mated: 234/205/192/173/154/177/96; Young *pie-1p*::*uev-2*: 182/192/162/125/116/97; Aged *pie-1p*::*uev-2* Mated: 140/149/136/127/126/95/102. Germlines distances were normalized to the start (0) and end (1) of pachytene based on DAPI morphology (see Methods). For analysis, the germline was divided into 7 bins encompassing the transition zone (Bin 1), early pachytene (Bins 2–3), mid pachytene (Bins 4–5), and late pachytene (Bins 6–7). The germline positions at which each bin start and end are marked on the X axis as vertical grey lines. Heat maps below each bin display the p values of pairwise comparisons of RAD-51 foci per nucleus counts within that bin. P values were calculated using Mann-Whitney U tests with Holm-Bonferroni correction for multiple comparisons. Blue tiles indicate significant differences (adjusted p value <0.05) and grey tiles indicate nonsignificant effects (adjusted p value >0.05). Young and aged mated wild type data is shared with Figs 1 and 6. Numerical data associated with this figure is presented in S1 Data.

young wildtype 1.6±2.2, aged wildtype 2.0±4.9, young *pie-1p*::*uev-2* 2.2±2.1 Mann-Whitney U test p<0.05; Bin 7 young wildtype 0.9±1.4, aged wildtype 0.5±1.9, young *pie-1p*::*uev-2* 1.0±1.3 Mann-Whitney U test p<0.05). Conversely, aged mated *pie-1p*::*uev-2* germlines maintained significantly fewer RAD-51 foci throughout late pachytene than young wildtype or young *pie-1p*::*uev-2* germlines (Fig 6B and 6C, Bin 6 young wildtype 1.6±2.2, aged wildtype 2.0±4.9, aged *pie-1p*::*uev-2* 0.5±0.9 Mann-Whitney U test p<0.05; Bin 7 young wildtype 0.9±1.4, aged wildtype 0.5±1.9, aged *pie-1p*::*uev-2* 0.1±0.3 Mann-Whitney U test p<0.05). Taken together, these data suggest that UEV-2 is not the sole regulator of DSB repair efficiency during *C. elegans* germline aging and appears to have age-dependent functions in regulating meiotic DSB accumulation and repair.

## Discussion

*C. elegans* germline function is impacted both by reproductive aging and sperm signals. Our study demonstrates that aged *C. elegans* germlines exhibit delayed DSB repair in mid-pachytene regardless of mated status, suggesting that deficiencies in germline DNA repair are a product of reproductive aging. We further find that sperm depletion, but not absence of sperm, reduces RAD-51 marked DSBs at early pachytene, suggesting that loss of signals from sperm downregulate DSB induction. Taken together, our study supports a model in which signals due to sperm depletion and reproductive aging operate in parallel to influence meiotic DSB induction and repair (Fig 7).

### Sperm depletion and DSB induction

Our data indicate that aged unmated germlines exhibit dramatically reduced RAD-51 foci in early pachytene (Figs 1 and 7). Previous work has similarly reported that unmated hermaphrodites induce fewer DSBs with age [8,10]. We find, however, that mating is sufficient to rescue RAD-51 foci accumulation at early pachytene in aged germlines. This effect in mated hermaphrodites may be due to cues from sperm-specific signals, seminal fluid components, or male pheromones, all of which impact hermaphrodite physiology [22,23,43]. While our study cannot distinguish between these male- and mating-dependent effects, it is notable that aged feminized *fog-2* germlines, which have never been exposed to sperm or males, do not exhibit reduced RAD-51 marked DSBs during aging. Further, we demonstrate that depletion of sperm from mating is sufficient to downregulate RAD-51 foci in old *fog-2* mutants. Thus, we suggest that DSB induction in aged germlines is primarily repressed by signals caused from sperm depletion rather than reproductive aging, mating-induced signaling, or exposure to males. Multiple mutants in *C. elegans* have been reported to exhibit age-dependent decline in meiotic DSB induction [30,44,45]. Our research raises the possibility that these proteins may mediate or respond to signals from sperm. Further, we observed a subtle but significant increase in RAD-51 foci in the transition zone of aged mated worms, suggesting that supplementation of

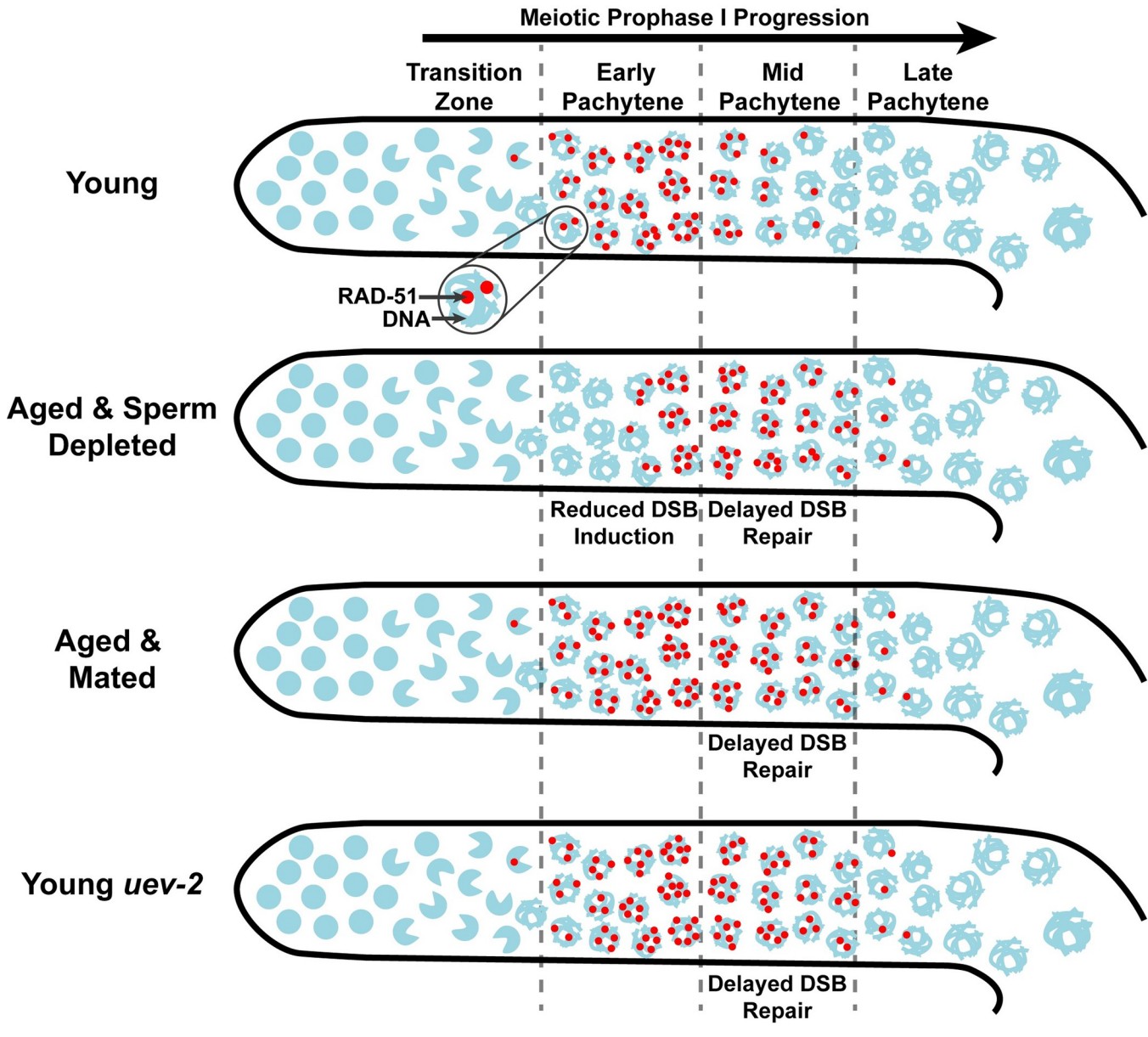

**Fig 7. Model of aging and sperm effects on DSB levels during germline aging.**

additional sperm, male sperm specific effects, or mating may also promote DSB formation in earlier meiotic stages.

Why might sperm depleted hermaphrodites downregulate germline DSB induction? Recent evidence has unveiled a potential transition in hermaphrodite gonad function following sperm depletion [46]. After all sperm have been utilized from the spermatheca, hermaphrodites continue to lay unfertilized oocytes and secrete intestine-derived yolk proteins in what has been suggested to be a form of 'primitive lactation' [46]. Thus, the reduction in DSBs induced in germ cells following sperm depletion may be a product of the hermaphrodite germline functionally changing from a reproductive organ to a system which produces food for offspring. Reduced DSB formation, then, may be indicative of the metabolic resources of the hermaphrodite being reallocated in favor of providing nutritional supplement for progeny.

Both reproductive aging and *C. elegans* yolk secretion are regulated by insulin/insulin-like growth factor signaling (IIS) [4,46]. IIS in *C. elegans* is mediated by the insulin receptor DAF-2, which promotes the phosphorylation of the FoxO transcription factor DAF-16, thereby repressing DAF-16 activity and sequestering it in the cytoplasm [47]. In the absence of DAF-2 signaling, DAF-16 localizes to the nucleus and acts with protein phosphatase 4 (PP4) to alter gene transcription and promote longevity [47,48]. Reduced DAF-2 signaling promotes reproductive longevity and represses yolk secretion following sperm depletion [4,5,46]. Sperm depletion, but not absence of sperm, reduces DAF-16 nuclear localization in the intestine, indicating that loss of sperm has tissue-nonautonomous effects on IIS [49]. Notably, a recent study found that PP4 also maintains DSB induction in the *C. elegans* germline by promoting the dephosphorylation of the Rec114 homolog DSB-1 [45]. From this evidence, we propose that the influence of sperm depletion on DSB induction may be mediated through the IIS pathway. Specifically, signals from sperm depletion may downregulate PP4 activity in addition to altering DAF-16 localization, thereby promoting the production and secretion of yolk proteins and reducing DSB induction via the phosphorylation of DSB-1. In summary, our work illuminates a regulatory mechanism specifically associated with sperm depletion which downregulates DSB induction in the *C. elegans* germline.

We further found that *spo-11(me44)* mutants do not exhibit increased DSBs with age, suggesting that the DSBs observed in aged gonads come from the endogenous meiotic machinery. This result contrasts with previous work done using aged *spo-11(ok79)* mutants, which incur SPO-11 independent DSBs [8]. The *spo-11(ok79)* allele is a 1227 bp deletion which removes the first 245 bp of the coding sequence and ablates the translation start site [2]. In contrast, *spo-11(me44)* is a glycine to alanine substitution mutation in a highly conserved residue and therefore encodes a protein which is likely translated but is functionally null [50]. As Spo11 is known to be required for localization of other pro-DSB proteins in budding yeast [51], it is possible that these other DSB induction proteins are differentially regulated during aging in the absence of SPO-11, contributing to the DSBs observed in aged *spo-11(ok79)* mutants. Thus, our data raises the possibility that the specific nature of *spo-11* mutation or background strain-specific effects influence the occurrence of exogenous DSBs in aged *spo-11* mutant germlines.

## DSB repair and *C. elegans* reproductive aging

Aged *C. elegans* germlines exhibit multiple DNA repair defects, including delays in recombination protein loading and increased engagement of error-prone repair mechanisms [8]. Both RAD-51 loading and error prone pathway engagement are regulated by DSB end resection [35], suggesting that differences in DSB repair during aging may be derived from defects at this DNA processing step. We demonstrate that the E2 enzyme variant UEV-2 is required for 'youthful' patterns of RAD-51 foci resolution during mid pachytene, indicating that a loss of UEV-2 or an age-related change in its function may underly the DNA repair defects in aged germlines. However, overexpression of UEV-2 is not sufficient to rescue persistent RAD-51 foci at mid pachytene in aged germlines and instead introduces DSB repair defects in young germlines. These results suggest that the specific levels of *uev-2* expression or the co-expression of other proteins may be important for the function of UEV-2 in DNA repair processes. Germ cell-specific sequencing suggests that *pie-1* is endogenously expressed at levels >200 fold higher than *uev-2* in a wild type context [52]. As our *uev-2* overexpression construct is driven by the *pie-1* promoter, the DSB repair phenotypes we observe in our *uev-2* overexpression strain are possibly related to inappropriate expression levels of UEV-2. Future experiments using weaker or even stronger germline promoters to drive *uev-2* expression may enable more

fine-scale dissection of how age-related changes in dosage of this gene influences aging phenotypes.

While the specific molecular functions of UEV-2 remain unknown, previous yeast two-hybrid assays have shown that UEV-2 may interact with BRC-1, the *C. elegans* BRCA1 homolog [53]. BRCA1 is an E3 ubiquitin ligase thought to regulate many DNA repair steps, including DSB resection [54]. Recent studies have demonstrated that BRC-1 is vital for preventing error prone repair in the *C. elegans* germline [55,56]. Given these results, we propose that UEV-2 may modulate BRC-1 activity in the germline to regulate resection of DSBs and promote efficient recombination. Under this model, overexpression of *uev-2* or loss of its function may cause hyper- or hypo-DSB resection respectively, and thus have a deleterious impact on the efficiency of recombination. RNAseq analysis of the aging transcriptome in worms 1, 10, and 20 days post-L4 has found that *uev-2* is upregulated with age [57]. As *uev-2* promotes germline DSB repair in long-reproductive *sma-2* mutants [4], this transcriptional pattern may indicate that *uev-2* is upregulated to compensate for defects in meiotic DSB repair processes which are accrued with age. Taken together, our work suggests that UEV-2 is involved in regulating efficient and 'youthful' meiotic DSB repair, thereby opening avenues to future work uncovering the specific roles this protein plays in meiosis.

## Supporting information

**S1 Fig. RAD-51 focus accumulation is delayed in aged unmated *rad-54* mutant germlines.** Violin and box plots depicting the number of RAD-51 foci in early pachytene germline nuclei of young (day 1 adult) and aged (day 4 adult) mated or unmated animals. Nuclei were quantified based on their positions in rows progressing through the germline on the distal proximal axis. Row 1 indicates the first row of nuclei in early pachytene. P values were calculated by pairwise Mann-Whitney U tests within each row of nuclei with Bonferroni correction for multiple comparisons. For simplicity, only significant (corrected p value ≤0.05) comparisons are displayed. Nuclei were scored from 6 germlines for each age and mating group (young, aged mated, aged unmated) derived from ≥2 experimental replicates. Each violin and boxplot represents an average of 29.5±3.3 nuclei (minimum 24 nuclei, maximum 37 nuclei, median 29 nuclei). Numerical data associated with this figure is presented in S2 Data.
(TIF)

**S2 Fig. DSBs in aged germlines are SPO-11 dependent.** A) Schemes used to isolate young (1 day post-L4) and aged (4 days post L4) worms for experiments. B-D) Representative whole gonad images of RAD-51 stained germlines from young and aged mated *spo-11(me44)* mutants. Top panels show merged images of both RAD-51 and DAPI, while lower panels show only RAD-51 staining in greyscale. Gonads are oriented with the distal mitotic tip on the left and the end of pachytene on the right. Gonads are outlined with grey dashed lines and scale bars represent 20μm.
(TIF)

**S3 Fig. Aged mated and unmated germlines maintain DSB-2 localization in early pachytene.** A) Representative images of germlines stained with DSB-2. Solid lines indicate the "DSB-2 zone", defined as the region of the germline in which >50% of nuclei are stained with DSB-2. Dashed lines extend from the end of the DSB-2 zone to the most proximal nucleus which has DSB-2 staining. Scale bars represent 20μm. B) Line plot representing the quantification of DSB-2 staining in young, aged mated, and aged unmated N2 hermaphrodite germlines. For specific maintenance schemes of these groups, see Fig 1A and Methods. Each horizontal line represents the portion of a single germline which contains DSB-2 positive nuclei. Solid lines

represent the "DSB-2 zone", while dashed lines extend to the most proximal germline position at which 1 or more nuclei is marked with DSB-2. C-D) Violin plots comparing the end of the DSB-2 zone and the final position of DSB-2 positive nuclei in young, aged mated, and aged unmated germlines. P values were calculated by Mann-Whitney U test with Bonferroni correction for multiple comparisons. Numerical data associated with this figure is presented in S7 Data.
(TIF)

**S4 Fig. A population of *fog-2* mutant oocytes exhibit reduced viability with parental age.** Bar plots representing the population brood viability of mated *fog-2* mutant females. In this experiment, *fog-2* females were maintained in the absence of males for 1–5 days post-L4 stage (X axis) and were then successively mated for three days. The survival rate of ovulated progeny is depicted for each successive day of mating 'First/Second/Third Day Mated' (see Materials and Methods). Error bars represent 95% Binomial confidence intervals. P values were calculated by Fisher's Exact Test. N values indicate the total number of live progeny and dead eggs scored. P values >0.05 are indicated as n.s. (not significant). Numerical data associated with this figure is presented in S3 Data.
(TIF)

**S5 Fig. The transition zone is reduced/absent in aged feminized germlines.** Representative images of fog-2(q71) feminized mutant germlines from animals 1, 3, or 4 days post-L4. The transition zone is marked with a solid blue line in germlines which have crescent shaped nuclei indicative of meiotic entry. Dashed lines indicate the regions of the germline presumably bridging the mitotic and meiotic germline in which crescent shaped 'transition zone' nuclei are absent. Gonads are oriented with the distal mitotic region on the left and the proximal meiotic region on the right. Scale bars represent 20μm.
(TIF)

**S6 Fig. Irradiated *fog-2* germlines exhibit high internuclear variance in DSB repair following irradiation.** A) Representative images of germlines from young and aged irradiated and unirradiated *fog-2(q71)* germlines. For specific maintenance schemes of each group, see Fig 3A. Scale bars represent 20μm. Grey numbered boxes indicate inset panels of nuclei displayed in panel B. B) Represented images of subsets of nuclei from germlines displayed in panel A. Numbers on images correspond to the grey boxes in panel A indicating the portion of the germline each image is derived from. Scale bars represent 5μm. C) Dot plots indicating the RAD-51 foci per nucleus in *fog-2(q71)* IR or No IR young and aged. Each point represents a single nucleus at a given germline position normalized by the premeiotic tip (0) to late pachytene (1) (see Methods). D) Variance in RAD-51 foci per nucleus calculated in a sliding window along the length of the germline where the width of the window is 0.1 germline distance units and the step size is 0.01 germline distance units. Average nuclei quantified in each bin ± standard deviation: Young *fog-2* No IR 141.6±26.6, Young *fog-2* IR 134.4±27.9, Old *fog-2* No IR 148.5±29.6, Old *fog-2* IR 142.4±30. Numerical data associated with this figure are presented in S1 Data.
(TIF)

**S7 Fig. Diagram of *uev-2(gk960600gk429008gk429009)* sequence structure.** Displayed is a scale cartoon of the *uev-2* locus where exons are displayed as boxes and intronic or noncoding upstream sequence is displayed as lines. The gk960600 allele deletes the translation start site and the 5' intron boundary of exon 2. This lesion generates a frameshift mutation and likely eliminates gene function. Additional point mutations gk429008 and 429009 cause single amino acid substitutions. Base pair distances are indicated relative to the translation initiation

site of Exon 1 of the *uev-2* coding sequence.
(TIF)

**S1 Data. Associated with Figs 1, 2, 3, 5, 6 and S6.** RAD-51 foci quantified in individual nuclei in wild type (N2), *uev-2*, *pie-1p::uev-2*, or *fog-2* germlines. Each row details a single quantified nucleus. Columns indicate the genotype of the animal (Genotype), its age in days post L4 (Days_post_L4), whether the animal was mated (Mated), the position of a nucleus along the length of the germline either normalized to the length of pachytene (Germline_Positon_Pachytenenormalized, see Methods) or normalized to the length of the germline (Germline_position, see Methods), the radiation dose applied to the worm if applicable (Treatment), and the number of RAD-51 foci associated with that nucleus (RAD51_foci).
(XLSX)

**S2 Data. Associated with S1 Fig.** RAD-51 foci quantified in individual nuclei in *rad-54* mutant germlines. Each row details a single quantified nucleus. Columns indicate the genotype of the animal (Genotype), its age in days post L4 (Days_post_L4), the row of the nucleus in the germline (Row), whether the animal was mated (Mated), and the number of RAD-51 foci (RAD51_foci).
(XLSX)

**S3 Data. Associated with S4 Fig.** Brood viability of *fog-2* animals aged 1–5 days post L4 and mated for three subsequent days (see Methods). Each row represents the brood of a single *fog-2* worm. Columns indicate the experimental replicate (Replicate), the ID of the *fog-2* female within that replicate and age group (PlateID), the age of the *fog-2* female in days post-L4 at the time of first mating (Age_when_mated), the day post mating from which the brood was scored (Day), the number of progeny which successfully hatched into larvae (Live), the number of unhatched dead eggs (Dead), the number of unfertilized oocytes (Unf), and whether live progeny were observed within the mated *fog-2* female (Matricidal Hatching).
(XLSX)

**S4 Data. Associated with Fig 4.** GFP::COSA-1 foci quantified in individual nuclei of *fog-2* female worms. The number of nuclei (nuclei) with a given number of GFP::COSA-1 foci (COSA1_Count) are grouped based on the age of the *fog-2* female in days post L4 (Age_postL4) and whether the *fog-2* female was mated (Mated, see Methods). The fraction of nuclei within a specific age and mating group which have a specified number of GFP::COSA-1 foci are detailed in the fraction_nuclei column.
(XLSX)

**S5 Data. Brood viability and incidence of males in the broods of wild type and uev-2 mutant hermaphrodites.** Each row represents the brood of a single scored parent hermaphrodite. Columns indicate the genotype of the parent (Genotype), the parent's ID (Plate), the number of unhatched dead embryos (Dead_Egg), the number of hatched hermaphrodite progeny (Herm), and the number of hatched male progeny (Male), and the total number of progeny scored (Total Progeny).
(XLSX)

**S6 Data. Number of DAPI staining bodies scored in diakinesis nuclei of wild type and uev-2 mutants hermaphrodites.** Each row represents a single nucleus scored. Columns indicate the genotype of the parent worm (Genotype) and the number of DAPI staining bodies observed in a nucleus (DAPIbodies).
(XLSX)

**S7 Data. Associated with S3 Fig.** Region of wild type young and aged germlines which is stained with DSB-2. Each row represents a single quantified germline. Columns indicate the age and mated status of the germline (Age; Young = 1 day post L4, Aged = 4 days post L4), the gonad number (GonadID), the beginning of DSB-2 staining in germline distance normalized to pachytene (DSB2_start, see methods), the end of DSB-2 staining in germline distance normalized to pachytene (DSB2_end, see methods), and the position of the most proximal DSB-2 staining nucleus in germline distance normalized to pachytene (DSB-2_straggler, see methods).
(XLSX)

## Acknowledgments

We thank the CGC (funded by National Institutes of Health (NIH) P40 OD010440) for strains. We thank C. Cahoon, A. Naftaly, and N. Kurhanewicz for thoughtful comments on this manuscript.

## Author Contributions

**Conceptualization:** Erik Toraason, Diana E. Libuda.

**Formal analysis:** Erik Toraason, Victoria L. Adler, Diana E. Libuda.

**Funding acquisition:** Diana E. Libuda.

**Investigation:** Erik Toraason, Victoria L. Adler.

**Methodology:** Erik Toraason, Diana E. Libuda.

**Project administration:** Diana E. Libuda.

**Resources:** Diana E. Libuda.

**Supervision:** Diana E. Libuda.

**Writing – original draft:** Erik Toraason.

**Writing – review & editing:** Erik Toraason, Victoria L. Adler, Diana E. Libuda.

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
