## [Decision Letter · Decision Letter 0]

30 Jun 2022

Dear Dr Libuda,

Thank you very much for submitting your Research Article entitled 'Aging and sperm signals alter DNA break formation and repair in the C. elegans germline' to PLOS Genetics.

The manuscript was fully evaluated at the editorial level and by independent peer reviewers. The reviewers appreciated the attention to an important problem, but raised some substantial concerns about the current manuscript. These include:

1) Performing experiments in the rad-54 mutant background to truly distinguish between effects on double strand break induction and repair, particularly in the aged, sperm-depleted hermaphrodites which show variations in RAD-51 appearance, as Reviewer 1 suggested.

2) Commenting on or performing experiments to show whether UVE-2 expression/localization is affected by age, as reviewer 1 suggested.

3) Performing experiments or addressing in your text the distinction between exposure to sperm vs. being mated to males, as both reviewers 2 and 3 suggested.

4) Addressing in your text why irradiation of aged control hermaphrodites (mated and unmated) was not used as a control for irradiated aged fog-2 animals or performing these experiments, as reviewer 3 suggested.

5) Explaining how many nuclei are excluded from analysis in their automated analysis of RAD-51 appearance and disappearance, as reviewer 2 suggested.

Based on the reviews, we will not be able to accept this version of the manuscript, but we would be willing to review a much-revised version. We cannot, of course, promise publication at that time.

If you decide to revise the manuscript for further consideration at PLOS Genetics, please aim to resubmit within the next 60 days, unless it will take extra time to address the concerns of the reviewers, in which case we would appreciate an expected resubmission date by email to plosgenetics@plos.org.

[LINK]

We are sorry that we cannot be more positive about your manuscript at this stage. Please do not hesitate to contact us if you have any concerns or questions.

Yours sincerely,

Needhi Bhalla

Guest Editor

PLOS Genetics

Gregory P. Copenhaver

Editor-in-Chief

PLOS Genetics

Reviewer's Responses to Questions

**Comments to the Authors:**

Reviewer #1: The manuscript by Toraason et al., characterizes the contribution of oocyte aging and sperm depletion on meiotic DSB formation and processing in the C. elegans germ line. Taking advantage of unmated and mated hermaphrodites and the obligate female fog-2 mutants, the authors carefully tease apart the impact of sperm depletion and aging on RAD-51 kinetics in the germ line. Additionally, the authors provide preliminary evidence that the E2 variant, UVE-2, contributes to the aging phenotype. For the most part, the data is very clear and well presented. The following should be addressed:

1. As the field is limited to RAD-51 as a readout of both DSB formation and processing, it is difficult to tease apart how sperm depletion and aging impact DSB formation vs. processing. I think it would be useful to examine total number of breaks by inactivation of RAD-54 under the different conditions. While I appreciate that this has the added complication of activating signaling pathways to maintain DSB formation, it would still provide important information about DSB formation vs. processing, which is central to many of the conclusions. I also note that while the authors were very careful in the results section to not specifically attribute RAD-51 foci to DSB formation, in the discussion they specifically equate DSB induction with RAD-51 foci.

2. UVE-2: The authors show that mutation of uve-2 leads to a “premature aging” phenotype with respect to RAD-51 foci. It would really strengthen the paper if the authors examined localization/expression of UVE-2 in the germ line over time – does UVE-2 expression decrease with age? I also think it will be very important to measure the extent of over-expression of UVE-2 from the pie-1 driven construct in comparison to young and old wild-type expression given that the effect observed was subtle.

3. In the discussion the authors suggest that the contradictory finding concerning SPO-11-dependence between this study and a previous study could be due to the nature of the SPO-11 allele and/or strain background differences. I recommend including the molecular details of the two alleles directly in the discussion.

4. I really like the figures with respect to RAD-51 and the thorough statistics presented. Just a couple of comments: In figure 1, wouldn’t n.d. be better as n.a. (not applicable)? In figure 5, the statistical comparison between young uev-2 and wt, should be blue as the p value is 0.043.

5. I had a hard time deciphering supplemental figure 3: if fog-2 was mated on the second day, why is there values for day 1 brood viability (etc)? I am obviously missing something and suggest that the authors clarify the experimental set up.

Reviewer #2: Review for PGENETICS-D-22-00650

Summary

In this manuscript, the competence of meiotic recombination (DNA break initiation and repair) in C. elegans oocytes was assessed as a function of organismal age and history of exposure to sperm, and found to correlate significantly with both factors in distinct ways. Depletion of sperm in organisms that once held sperm was shown to lead to reduction of break initiation, while age alone irrespective of sperm history led to delays in break repair. Since age and sperm depletion are normally precisely correlated in C. elegans hermaphrodites, the attempt to decouple these two causes was a great idea and has succeeded in bringing new understanding to the causes of decreasing oocyte quality. The manuscript leads naturally into future work where the mechanisms of how age, sperm depletion, and other factors could affect DNA repair may be elucidated.

Critiques

1

The data are of high quality, in general well-controlled and in sufficient biological replicates, and the statistics appear properly done and corrected for multiple comparisons where appropriate. The method of straightening gonads and measuring foci in sliding windows of normalized distance, rather than the standard practice of going straight to bins, allows a more comprehensive look at the course of foci appearance and disappearance, and should be more widely adopted. One concern I have with the method as described is the reliance on automation and in particular the elimination of nuclei that overlap others (Methods 260-261). It would help the appraisal of the data to be told how many nuclei are typically thus eliminated, and whether there is any possibility that elimination is non-random with respect to RAD-51 focus number. Even if this information is available in the first author's earlier method publication, it would be good to make it explicit here as well. It would defeat the purpose of automation to do this for every single gonad, but a statement clarifying this elimination step and the possibility for bias would be useful.

2

A question/concern about the nature of the meiotic spatiotemporal gradient in *fog-2* animals: The IR experiment showing defects in DNA repair in aged *fog-2* feminized germlines examines DNA breaks that have not been repaired even after 2 days. Normally the time from S phase to pachytene is 24-48 hours (Jaramillo-Lambert et al.) so the expectation for wild-type germlines would be that the pre-pachytene stages should not show IR-induced damage after 48 hours. The introduction (385-) explains that "Due to the absence of signaling from sperm in *fog-2* mutants, both germline stem cell proliferation and meiotic progression are halted, such that meiotic oocytes are held within the gonad" which explains why the damage is still visible. But in Results (457–) it is stated "aged irradiated germlines maintained a median elevation of ~6-10 RAD-51 foci per nucleus (germline position 0.5-1.0, Figure 3C). This result indicates that aged fog-2 germlines exhibit DNA repair defects specifically in nuclei at later stages of meiotic prophase I.", here explicitly identifying late germline position with late stages of meiotic prophase I. My question (after all this setup) is, to what extent do we know that proximal position in the germline reliably indicates "late meiotic prophase", when all cells in the gonad are old enough to be at least in mid-pachytene? As you show that the transition zone is missing (Sup. Fig.4), it seems that meiosis does progress in these halted cells. So, can the higher RAD-51 foci over background in irradiated *fog-2* aged germlines be properly attributed to the temporal stage of prophase, or could it perhaps be an effect only of spatial position? This experiment just made me think about the assumptions involved in doing the usual spatiotemporal analysis an essentially static gonad, so I think it would be good to guide the reader through the logic in a little more detail.

3

The conclusion that sperm *depletion*, not sperm absence or age, is causing a reduction in break initiation in early pachytene receives support from (1) the difference between (aged and sperm-depleted wild-type) versus (aged and mated wild-type) animals, and (2) the difference between (aged and unmated wild-type) and (aged and unmated *fog-2*) animals (e.g. discussion 567-569), the latter having never encountered sperm. The evidence would be stronger still if *mated but sperm-depleted aged fog-2* animals could be examined, since the prediction is that they would show the same DSB initiation defects as wild-type sperm-depleted animals. Technical considerations may render this experiment unfeasible, and this result is not necessary to accept the conclusion, but for the manuscript's *fog-2* results to count as evidence I think it must be assumed that *fog-2* germlines would have shown reduced RAD-51 foci had they been transiently exposed to sperm.

Minor comments:

Lines 338-340 "During early pachytene, the amount of RAD-51 foci per nucleus was similar between aged mated germlines and young germlines…Young germlines, however, accumulated a higher total number of RAD-51 foci per nucleus…" the second part here should contain a reference to the later stage it refers to, otherwise it may confuse the reader (is it similar or higher?)

Fig. 1, 5, and 6 have mislabeled panels (A,B,B instead of A,B,C).

Fig. 2, the "RAD-51 zone" is defined once in the text and twice in the legend with slightly different wording each time; although the different descriptions are mostly consistent it would be better to define it only once.

Fig. 3, panel B shows mean focus number per nucleus, consistent with Figs. 1,2,5, and 6, but panel C (the subtraction between +/-IR) uses the median; if this is necessary, it should be given an explanation.

Fig. 3, the nomenclature for young vs. aged is changed somewhat from Fig. 2 ("young" Fig. 2 animals are dissected at 1 day post-L4, while "young" Fig. 3 animals are dissected at 3 days) and could lead to confusion; a change in naming might help, but if no alternative seems better, then just explaining this in the Figure legend would suffice.

Fig. 6: it should be stated in the legends that the data for Young Wildtype and Aged Wildtype Mated are the same as Fig. 5.

Reviewer #3: Toraason et al., investigated the influence of two factors, sperm and reproductive aging, on oocyte quality in C. elegans. They specifically analyzed double-strand breaks (DSBs) induction and repair during meiotic prophase I in oocytes and how they are affected by aging, sperm and their combination. They dissected these factors by taking advantage of C. elegans hermaphrodites (WT) that produce oocytes and sperm, and obligate female mutants (fog-2) that produce only oocytes. Experimental design also accommodated the difference between complete lack of sperm exposure vs sperm depletion. The WT and obligate female mutants were used to investigate the role of sperm and aging. Authors concluded that sperm depletion instead of the complete absence of sperm downregulates DSBs suggesting that loss of signals from sperm causes deficiencies in DSBs induction and repair. Authors also confirm that reproductive aging is the main factor responsible for the DNA repair deficiency in the aged germline. In the second part of the manuscript authors show that the E2 ligase variant UEV-2 which has previously been implicated in DSB repair efficiency is important for DSBs in young germlines and that aging may affect its efficacy. Overall, the experiments that are reported are well executed and represented. However, some analyses seem incomplete as listed in major points below (for example radiation experiment is only done in fog-2 germlines which means only in the absence of sperm). Missing experiments are needed to strengthen the manuscript unless authors provide explanation to why they were not included or would not be informative. The notion that sperm signals are regulating oogenesis is known but the authors designed an original approach to dissect it genetically. Results will have a high to moderate importance on reproductive research in C. elegans. However, the relevance for human reproductive aging research is not clear. Two aspects of this model system are not relevant for human: 1) sperm has no effect on mammalian meiotic recombination; 2) Meiotic recombination in oocytes takes place in utero and aging affects meiotically arrested oocytes and their ability to repair DSBs. Authors did not investigate oocytes arrested in diakinesis or mature oocytes which are equivalent to human oocytes. To increase the importance and relevance, authors could provide stronger explanation how their findings inform aging research in other system. Minor and major points for authors to consider are listed below.

Results:

Minor:

General comment: it would be helpful if authors provided results as numbers in the text (i.e. number of foci quantified) rather than always referring reader to a figure. For example:

• Line 342-346: “ were greatly decreased throughout early pachytene as compared to both young and aged mated germlines (Figure 1B-C, Bins 2-3 Mann-Whitney U test p<0.001).

• Line 469-470 : “with a subtle but significant increase” please provide the values in text.

Major:

• Lines 377-379: Authors conclude that “the extent of DSB-2 marked pachytene nuclei is influenced both by aging and by the absence of sperm” giving both factors equal weight while in fact the effect of age was “subtle”. I would recommend rephrasing this conclusion to represent the data.

• Line 380-433: Why were fog-2 germlines not compared to N2 wildtype germlines? To determine the impact of sperm production n young and old germlines I would expect comparison between young WT vs fog-2, old WT mated vs fog-2 old mated and old WT unmated vs fog-2 old unmated. Provide explanation why these comparisons were not made.

• Lines 432-433: the model should be validated by analyzing DSB induction in fog-2 mutants mated to males

• Lines 447-449: Is this sensitive subpopulation specific to fog-2 mutant or also exist in WT? Again, explain lack of comparison with WT irradiated.

• Lines 460-461: Without comparison with WT irradiated germlines I’, not sure authors can conclude that“ that DNA repair efficiency is altered in aging germlines independent of any signals from sperm” because analysis was performed only in females without sperm. If the conclusion is valid, then similar results should be seen in aged hermaphrodites mated and unmated after IR.

• Lines 479-480: Similarly, can the CO formation be analyzed in fog-2 mutants depleted of sperm to validate the conclusion?

• The analysis of uev-2 mutants does not include analysis of crossovers using COSA marker. It would improve the manuscript if impact of aged-like-DSB repair was tested for consequences in Crossovers and oocyte quality.

Discussion:

Minor:

• Line560-561: Refer to actual data figure or specify that this is referring to a model.

Major:

• Line569-571: Without data from fog-2 mated females I don’t think authors can conclude that mating has no effect. Is there a male mutant that doesn’t have sperm but still mates with females? This could test how mating may affect DSB induction .

• Line 572-573: This is also different than what happens in human oocytes. Comment of this in the context of human relevance.

• Line 578-591: Authors provide speculative hypothesis why these changes in DSB induction may be occurring but don’t discuss how besides implicating insulin/insulin-like growth factor. Why specifically DSB? Why only reduction/delay and not complete cessation?

• Comment on early induction of DSBs in Aged mated compared to young (figure 1, Bin 1). In other words, comment on the significance in aged mated to both aged unmated and young. Is it due to the presence of exogenous sperm? As male sperm is known to outcompete hermaphrodite sperm so maybe male sperm may also have a “strong signal” for DSB induction?

Abstract:

Minor:

• Line 53 input short name “(C. elegans)-

• Line 59: … meiotic prophase I progression

Author Summary

Minor:

• Line 75: input short name “(C. elegans)

Methods:

Minor:

• Lines 163 and 161: N2 (wild type) repeated

• Lines 166-172: Make another section may be labeled “experimental assay” and that can also include the fog-2 Brood viability assay

• Lines 222, 290: are “worms” needed after C. elegans?

• Lines 250-253 Provide suppliers and catalogue numbers for commercial antibodies.

• Mann-Whitney U test is not mentioned in statistical methods description.

Figures:

Figure 1:

• Define n.d

• Line plot mislabeled as panel B

Figure 5 and 6:

• Line plot mislabeled as panel B

**Have all data underlying the figures and results presented in the manuscript been provided?**

Reviewer #1: Yes

Reviewer #2: **No: **I could not find spreadsheets in the main or supplemental figures containing numerical data that the RAD-51 focus graphs were based on.

Reviewer #3: Yes

PLOS authors have the option to publish the peer review history of their article (what does this mean?). If published, this will include your full peer review and any attached files.

Reviewer #1: No

Reviewer #2: **Yes: **Peter Carlton

Reviewer #3: No

---

## [Decision Letter · Decision Letter 1]

10 Oct 2022

Dear Dr Libuda,

Thank you very much for resubmitting your Research Article entitled 'Aging and sperm signals alter DNA break formation and repair in the C. elegans germline' to PLOS Genetics.

As you can see, the reviewers appreciate the changes you made in response to their reviews. However, Reviewer 1 would still like you to address the level of UEV-2 over expression from the pie-1 promoter, to contextualize the observed phenotype. We therefore ask you to modify the manuscript according to the review recommendation. 

Yours sincerely,

Needhi Bhalla

Guest Editor

PLOS Genetics

Gregory P. Copenhaver

Editor-in-Chief

PLOS Genetics

Reviewer's Responses to Questions

**Comments to the Authors:**

Reviewer #1: The revised manuscript by Toraason et al., has largely addressed the previous reviews. The work provides an important contribution to the field and insight into how aging and signals from sperm influence different aspects of meiotic recombination. Two issues remain:

1) The authors refer to UEV-2 as a variant E2 ubiquitin ligase – based on all the information I can find (which wasn’t very much) UEV-2 is an E2 conjugating enzyme.

2) A point that wasn’t addressed was the extent of overexpression of UEV-2 driven by the pie-1 promoter. I still think this is important– is it 2-fold, 10-fold, 100-fold overexpressed? This is relevant to interpreting the observed phenotype.

Reviewer #2: The authors have satisfied all my concerns through the amendments in this revision; in particular it is nice that they were able to do the "mated, sperm-depleted fog-2" experiment which two reviewers suggested. The results turned out to support their model, which places the findings on an even more solid footing.

Reviewer #3: I am satisfied with the revised version and have no further comments.

**Have all data underlying the figures and results presented in the manuscript been provided?**

Reviewer #1: Yes

Reviewer #2: Yes

Reviewer #3: Yes

PLOS authors have the option to publish the peer review history of their article (what does this mean?). If published, this will include your full peer review and any attached files.

Reviewer #1: No

Reviewer #2: **Yes: **Peter Carlton

Reviewer #3: No

---

## [Editor Report · Decision Letter 2]

21 Oct 2022

Dear Dr Libuda,

Thank you for addressing the comments of Reviewer 1. We are pleased to inform you that your manuscript entitled "Aging and sperm signals alter DNA break formation and repair in the C. elegans germline" has been editorially accepted for publication in PLOS Genetics. Congratulations!

Yours sincerely,

Needhi Bhalla

Guest Editor

PLOS Genetics

Gregory P. Copenhaver

Editor-in-Chief

PLOS Genetics

Comments from the reviewers (if applicable):

**Data Deposition**

http://datadryad.org/submit?journalID=pgenetics&manu=PGENETICS-D-22-00650R2

**Press Queries**

---

## [Editor Report · Acceptance letter]

1 Nov 2022

PGENETICS-D-22-00650R2 

Aging and sperm signals alter DNA break formation and repair in the C. elegans germline 

Dear Dr Libuda, 

We are pleased to inform you that your manuscript entitled "Aging and sperm signals alter DNA break formation and repair in the C. elegans germline" has been formally accepted for publication in PLOS Genetics! Your manuscript is now with our production department and you will be notified of the publication date in due course.

With kind regards,

Zsofia Freund

PLOS Genetics

On behalf of:
